# Lola regulates *Drosophila* adult midgut homeostasis via non-canonical hippo signaling

Xue Hao[1], Shimin Wang[1], Yi Lu[1], Wentao Yu[1], Pengyue Li[1], Dan Jiang[1], Tong Guo[1], Mengjie Li[2], Jinhui Li[1], Jinjin Xu[1], Wenqing Wu[1], Margaret S Ho[3]*, Lei Zhang[1,3]*

[1]State Key Laboratory of Cell Biology, CAS Center for Excellence in Molecular Cell Science, Shanghai Institute of Biochemistry and Cell Biology, Chinese Academy of Sciences, University of Chinese Academy of Sciences, Shanghai, China; [2]State Key Laboratory of Microbial Metabolism, School of Life Sciences and Biotechnology, The Joint International Research Laboratory of Metabolic and Developmental Sciences, Shanghai Jiao Tong University, Shanghai, China; [3]School of Life Science and Technology, ShanghaiTech University, Shanghai, China

**Abstract** Tissue homeostasis and regeneration in the *Drosophila* midgut is regulated by a diverse array of signaling pathways including the Hippo pathway. Hippo signaling restricts intestinal stem cell (ISC) proliferation by sequestering the transcription co-factor Yorkie (Yki) in the cytoplasm, a factor required for rapid ISC proliferation under injury-induced regeneration. Nonetheless, the mechanism of Hippo-mediated midgut homeostasis and whether canonical Hippo signaling is involved in ISC basal proliferation are less characterized. Here we identify Lola as a transcription factor acting downstream of Hippo signaling to restrict ISC proliferation in a Yki-independent manner. Not only that Lola interacts with and is stabilized by the Hippo signaling core kinase Warts (Wts), Lola rescues the enhanced ISC proliferation upon Wts depletion via suppressing *Dref* and *SkpA* expressions. Our findings reveal that Lola is a non-canonical Hippo signaling component in regulating midgut homeostasis, providing insights on the mechanism of tissue maintenance and intestinal function.

*For correspondence:
margareth@shanghaitech.edu.cn (MSH);
rayzhang@sibcb.ac.cn (LZ)

Competing interests: The authors declare that no competing interests exist.

## Introduction

Maintenance of tissue homeostasis, as in the intestinal epithelium, is under complex regulation so to achieve a dynamic balance in terms of the rate for cell turnover (*Guo et al., 2016*). Uncontrolled tissue regeneration leads to intestinal malignancies such as colorectal cancer (*Terzić et al., 2010*), whereas a steady and homeostatic condition is preferable to maintain the integrity of the intestinal tissue for its normal function such as food digestion and nutrient absorption (*Casali and Batlle, 2009*; *Guo et al., 2016*). The *Drosophila* adult midgut, functionally equivalent to the mammalian small intestine, consists of a single epithelial layer where mature cell types differentiate apical-basally from the intestinal stem cells (ISCs) scattered along the basal side (*Jiang et al., 2016*). ISCs undergo asymmetric divisions that give rise to a renewable ISC and a non-dividing immature enteroblast (EB), which further differentiates into either an absorptive enterocyte (EC) or a secretory enteroendocrine (ee) cell (*Micchelli and Perrimon, 2006*; *Ohlstein and Spradling, 2006*). Previous studies have shown that both ISCs and EBs, commonly referred as midgut precursors, express the Snail/Slug family transcription factor *escargot* (*Micchelli and Perrimon, 2006*). Whereas ISCs are marked by the Notch (N) ligand Delta (Dl) (*Ohlstein and Spradling, 2007*), EBs can be labeled by a reporter of N signaling, *Su(H)Gbe-lacZ* (*Su(H)-Z*), due to the activation by neighboring ISCs (*Micchelli and*

*Perrimon, 2006*; *Ohlstein and Spradling, 2006*). Terminally differentiated ECs, labeled by the class II POU domain transcription factor Pdm1 or Brush Border Myosin (MyoIA), acquire their large size and polyploid nuclei via endoreplication (*Jiang et al., 2009*; *Lee et al., 2009*). On the other hand, ee cells with small nuclei are specifically stained by Prospero (Pros) (*Micchelli and Perrimon, 2006*; *Ohlstein and Spradling, 2006*; *Singh et al., 2012*). These differential patterns in gene expression allow the identification of distinct cell types in the midgut, and provides strategic means to analyze the features and properties of adult stem cells.

Midgut homeostasis is maintained via a basal level of ISC turnover to replace the cells loss during normal gut function (*Antonello et al., 2015*; *Jiang et al., 2011*). Upon stress or injury, however, ISCs undergo rapid proliferation in order to replenish enough cells in a limited time for regeneration (*Amcheslavsky et al., 2009*; *Buchon et al., 2010*; *Buchon et al., 2009b*). These processes are regulated by a number of signaling pathways such as N (*Ohlstein and Spradling, 2007*), Wingless (*Lin et al., 2008*), JAK-STAT (*Beebe et al., 2010*; *Jiang et al., 2009*) and EGFR (*Buchon et al., 2010*; *Jiang et al., 2011*). Particularly, Hippo signaling has been shown to play pivotal roles in both *Drosophila* midgut homeostasis and regeneration via cell-autonomous and non-cell-autonomous mechanisms (*Karpowicz et al., 2010*; *Ren et al., 2010*; *Shaw et al., 2010*; *Staley and Irvine, 2010*). As an evolutionarily conserved pathway, Hippo signaling controls organ size by balancing cell proliferation and death (*Yin and Zhang, 2011*). The pathway consists of a core kinase cascade in which Hippo (Hpo) kinase phosphorylates and activates Warts (Wts) kinase via interaction with the scaffold protein Salvador (Sav). Subsequently, Wts interacts with Mob as tumor suppressor (Mats) to trigger phosphorylation of the transcription coactivator Yorkie (Yki), blocking its translocation to form a complex with the transcription factor Scalloped (Sd) in the nucleus, thus inhibiting downstream signal transduction (*Goulev et al., 2008*; *Harvey et al., 2003*; *Huang et al., 2005*; *Justice et al., 1995*; *Oh and Irvine, 2008*; *Pantalacci et al., 2003*; *Udan et al., 2003*; *Wu et al., 2003*; *Xu et al., 1995*). Despite that Hippo signaling mainly transduces via triggering Wts phosphorylation (*Udan et al., 2003*; *Wu et al., 2003*), previous studies indicate that some upstream components regulate the Hippo signaling activity by controlling Wts protein levels. The atypical cadherin Fat (Ft) (*Cho et al., 2006*), the atypical myosin Dachs (D) together with the LIM domain protein Zyxin (Zyx) (*Rauskolb et al., 2011*), and the tumor suppressor gene Scribble (Scrib) (*Verghese et al., 2012*) function as Hippo components via regulating Wts protein stability. During midgut homeostasis, Hippo signaling restricts ISC proliferation by sequestering Yki in the cytoplasm, thereby deactivating downstream signaling. Inactivation of Hpo or Wts leads to enhanced ISC proliferation, same as *yki* overexpression which activates EGFR and JAK-STAT pathways (in ECs, non-cell-autonomously) or promotes expression of target genes such as *bantam* microRNA (in ISCs, cell-autonomously) (*Houtz et al., 2017*; *Huang et al., 2014*; *Nolo et al., 2006*; *Ren et al., 2010*; *Shaw et al., 2010*; *Staley and Irvine, 2010*; *Thompson and Cohen, 2006*). In addition, the Yki-Sd complex is considered as the major mediator for injury-induced midgut regeneration, as loss of Yki in either ISCs or ECs blocks DSS- or infection-stimulated ISC proliferation, respectively (*Cai et al., 2010*; *Karpowicz et al., 2010*; *Ren et al., 2010*). Despite abundant evidence underlying the significance of Hippo signaling during midgut homeostasis and regeneration, neither Yki nor Sd depletion causes defects in ISC basal proliferation, raising the controversies whether the Yki-Sd complex truly executes downstream Hippo signaling during midgut homeostasis as it does in regeneration. *longitudinals lacking* (*lola*) is a transcription factor implicated in axon growth and guidance (*Crowner et al., 2002*; *Giniger et al., 1994*), ovary cell apoptosis (*Bass et al., 2007*), germline/neuron stem cell differentiation and maintenance (*Davies et al., 2013*; *Silva et al., 2016*; *Southall et al., 2014*) and wing margin development (*Bass et al., 2007*; *Krupp et al., 2005*). It also cooperates with Dl to induce the formation of metastatic tumors and regulates cell fate in *Drosophila* eye by antagonizing N signaling (*Ferres-Marco et al., 2006*; *Zheng and Carthew, 2008*). Despite the presence of more than 20 alternatively-spliced isoforms, the protein product of *lola* contains a N-terminal BTB or POZ domain and one or two C-terminal zinc finger motifs, indicating that Lola is a Cullin-3 (Cul3)-based substrate adaptor that also binds DNA (*Furukawa et al., 2003*; *Geyer et al., 2003*; *Goeke et al., 2003*; *Horiuchi et al., 2003*; *Ohsako et al., 2003*; *Pintard et al., 2004*). In the present study, we identify Lola as a transcription factor acting downstream of Hippo signaling to restrict ISC proliferation and regulate midgut homeostasis. Our results show that the Hippo signaling pathway component Wts physically interacts with Lola and regulates its stability. Whereas reduced *yki* expression does not rescue the enhanced ISC proliferation induced by Wts depletion, Lola genetically interacts with Wts

and exhibits a suppression effect on the overgrowth. Wts-mediated Lola stability provides a means for Hippo signaling to regulate midgut homeostasis in a manner independent of the canonical Yki-Sd complex. Furthermore, Lola restricts ISC proliferation by suppressing downstream *Dref* and *SkpA* expression levels. Taken together, our findings reveal that Lola is a novel signaling effector in regulating ISC basal proliferation and midgut homeostasis via its transcriptional activity and interaction with a central Hippo signaling component.

## Results

### Yki is dispensable for Wts-mediated ISC proliferation in *Drosophila* midgut

Given the importance of Hippo signaling in regulating *Drosophila* adult midgut homeostasis, it is rather surprising that lacking the Hippo effector Yki leads to no defects in ISC basal proliferation (*Karpowicz et al., 2010*; *Ren et al., 2010*; *Shaw et al., 2010*; *Staley and Irvine, 2010*). To first confirm if Hippo signaling regulates midgut homeostasis via the Yki-Sd canonical signaling, *escargot-Gal4* (*esg-Gal4*) that drives gene expression in ISCs and EBs was employed in combination with a temperature-sensitive Gal4 repressor *tub-Gal80$^{ts}$* to restrict the time of expression (*esg$^{ts}$*) (*Micchelli and Perrimon, 2006*). As shown in *Figure 1A–A'*, ISCs and EBs were labeled with *esg$^{ts}$*-driven GFP expression (esg-GFP). Expression of *wts* RNAi (*esg$^{ts}$ > wts* RNAi) resulted in an increase in the number of GFP$^+$ cells in the posterior part of the midgut, indicating that reduced *wts* expression leads to an expansion of the precursor cell population (*Figure 1B–B'* and *Figure 1—figure supplement 1A*). Consistent with previous results, reduced *yki* expression (*esg$^{ts}$ > yki* RNAi) does not lead to obvious difference in the number of GFP$^+$ cells, suggesting that Yki is not required for overall midgut homeostasis (*Figure 1C–C'*). Immunostaining with antibodies against phospho-Histone 3 (p-H3) that marks mitotic cells derived from ISCs indicated an obvious increase in p-H3$^+$ cell number when downregulating *wts* expression in ISCs and EBs (*esg$^{ts}$ > wts* RNAi, *Figure 1E*), suggesting that the expansion in the precursor cell population in *Figure 1B* is due to enhanced ISC proliferation. Unexpectedly, co-expression of *yki* and *wts* RNAi suppressed neither the growth of precursor cell population nor the enhanced proliferation induced by *wts* RNAi, suggesting that Wts mediates ISC proliferation independently of Yki (*Figure 1D–E* and *Figure 1—figure supplement 1A–F*).

Next, GFP$^+$ clones of *wts* mutant allele *wts$^{x1}$* were generated using the mosaic analysis with a repressible cell marker (MARCM) system (*Lee and Luo, 2001*). Interestingly, the number of Dl-labeled ISCs was increased in *wts$^{x1}$* clones, indicating that ISCs are over proliferative (*Figure 1F–G' and I*). Consistent to our *wts* RNAi results, simultaneous expression of *yki* RNAi in these clones did not restore the elevated Dl$^+$ cell number (*Figure 1H–I*), suggesting that the reduced *yki* expression does not block *wts$^{x1}$*-induced ISC proliferation. These results are inconsistent with previous findings that Yki works downstream of Wts and regulates the output of canonical Hippo signaling, raising concerns whether different mechanisms exist between gut and other tissues, or the *yki* RNAi line is not efficient enough in downregulating *yki* expression.

To address the inconsistency, we first sought to determine if Wts and Yki work in a tissue-specific manner. Using *Drosophila* wing as a model, expression of *wts* RNAi by *MS1096-Gal4* resulted in larger wing size, a phenotype rescued by *yki* RNAi co-expression. These results suggest that Yki functions together with Wts in controlling the wing size, a mechanism that differs from gut homeostasis (*Figure 1—figure supplement 1G–K*). These results are consistent with previous findings and indicate that *yki* RNAi is effective in downregulating *yki* expression. To further support our hypothesis, a separate MARCM clone approach independent of *yki* RNAi was employed using *yki* or *sd* mutant allele *yki$^{B5}$* or *sd$^{ΔB1}$*. Clones were induced in parallel in adult flies raised at 25 ℃ for 2 or 4 days after induction. Clone sizes were compared by measuring the number of GFP$^+$ cells per clone. Under this condition, we found that clones expressing *wts* RNAi were bigger than the control, and the size was not or only slightly affected by the presence of *yki$^{B5}$* or *sd$^{ΔB1}$*, respectively (*Figure 1J–N* and *Figure 1—figure supplement 1L–P*). Taken together, these results indicate a non-canonical mechanism regulating midgut homeostasis by Hippo signaling. The regulatory mechanism depends on Wts function, yet is independent of the Yki-Sd transcriptional complex in *Drosophila* midgut.

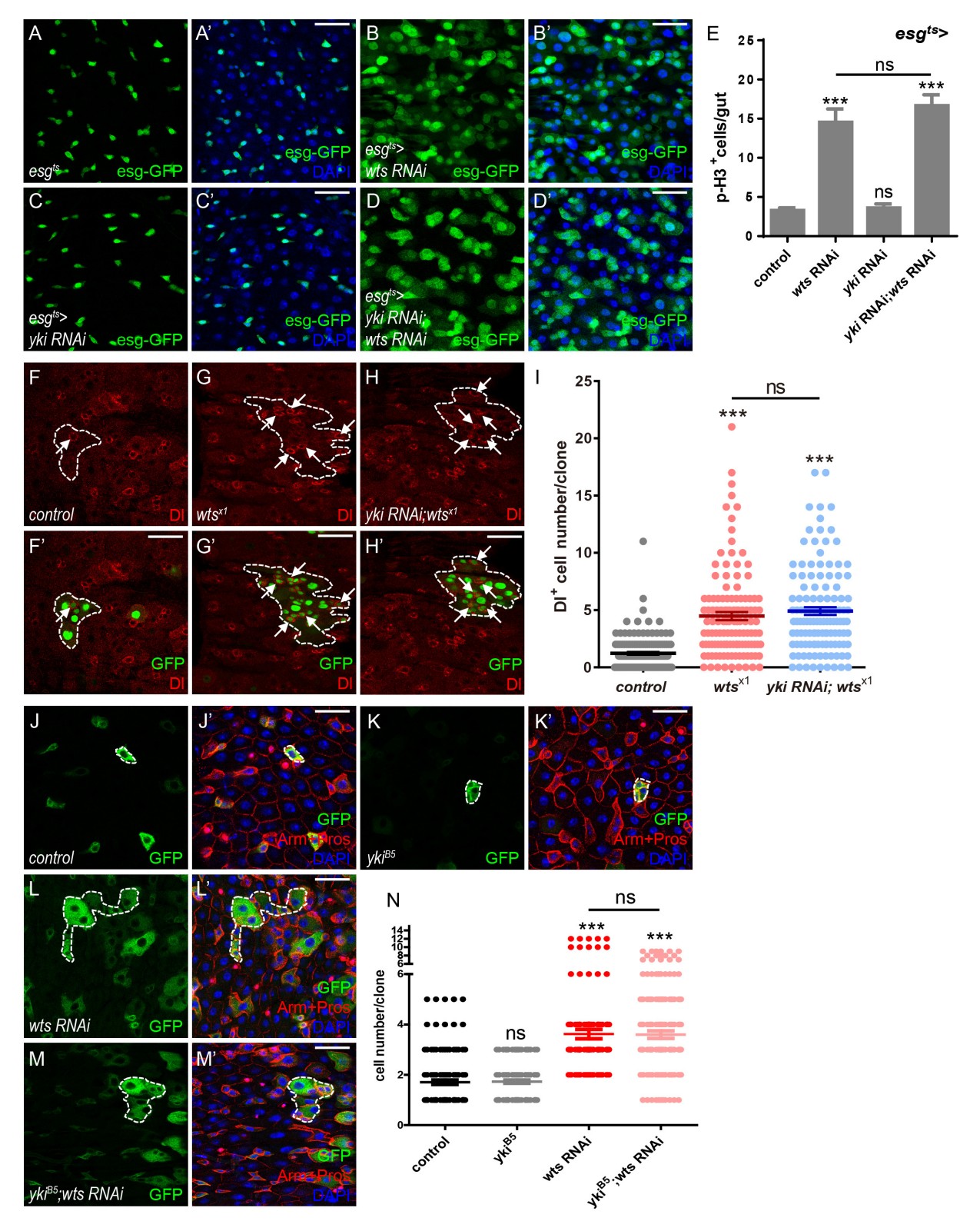

**Figure 1.** Yki is dispensable for Wts-mediated ISC proliferation in *Drosophila* midgut. (A–D') Representative images of *Drosophila* adult midguts of *esg-Gal4; tubGal80$^{ts}$* (*esg$^{ts}$*) (A–A'), *esg$^{ts}$ >wts* RNAi (B–B'), *esg$^{ts}$ >yki* RNAi (C–C'), and *esg$^{ts}$ >yki* RNAi; *wts* RNAi (D–D'). Midguts were dissected and immunostained with DAPI (nuclei, blue). ISCs and EBs are marked with esg-GFP (green). Merged images are shown in A', B', C', and D' (green and blue). (E) Quantifications of the p-H3$^+$ cell number in adult midguts with the indicated genotypes in A-D (n = 20, 21, 22, 16). Note a significant increase

*Figure 1 continued on next page*

*Figure 1 continued*

in the p-H3$^+$ cell number when *wts* RNAi is expressed (***p<0.001). Co-expression of *yki* and *wts* RNAi does not suppress the increase (ns p>0.05). (F–H') Representative images of *Drosophila* adult midguts containing GFP positive MARCM clones of control (F–F'), *wts*$^{x1}$ (G–G') and *wts*$^{x1}$ in the presence of *yki* RNAi (H–H'). Midguts were dissected and immunostained with antibodies against Dl (red) 4 days after clone induction. Areas enclosed by the dashed lines indicate clone regions. White arrows indicate ISCs marked by Dl. Merged images are shown in F', G' and H' (green and red). (I) Quantifications of the Dl$^+$ cell number per clone in adult midguts from the indicated genotypes in F-H. Average of 124–152 clones from 10 midguts for each genotype were quantified. (J–M') Representative images of *Drosophila* adult midguts containing GFP positive MARCM clones of control (J–J'), *yki*$^{B5}$ (K–K'), *wts* RNAi (L–L'), and *yki*$^{B5}$ in the presence of *wts* RNAi (M–M'). Midguts were dissected 2 days after clone induction and immunostained with antibodies against Arm and Pros (red) and DAPI (nuclei, blue). Areas enclosed by the dashed lines indicate respective clone size. Merged images are shown in J', K', L', and M' (green, blue, and red). (N) Quantifications of the cell number per clone in adult midguts from the indicated genotypes in H-K. Note that *yki*$^{B5}$ does not restore the large clone size induced by *wts* RNAi expression (red). Average of 122–176 clones from 10 midguts for each genotype were quantified. Scale bars: 30 μm. Data are shown as mean ± SEM. P-values of significance (indicated with asterisks, ns no significance p>0.05, *p<0.05, **p<0.01, and ***p<0.001) are calculated by Student's T-test. Confocal images were taken from the basal layer of the midgut where ISCs can be clearly visualized. Single layer image is shown.

The online version of this article includes the following source data and figure supplement(s) for figure 1:

**Source data 1.** Source data for *Figure 1E*, *Figure 1I*, *Figure 1N*, *Figure 1—figure supplement 1F*, *Figure 1—figure supplement 1K*, and *Figure 1—figure supplement 1P*.

**Figure supplement 1.** Yki is dispensable for Wts-mediated ISC proliferation in *Drosophila* midgut (A)Relative *wts* mRNA levels were analyzed by real-time PCR.

## Lola is required for ISC proliferation and midgut homeostasis

Based on our results, we propose that Wts-mediated Hippo signaling regulates midgut homeostasis independently of Yki. To identify factors that might act downstream of Wts in mediating midgut homeostasis, a genetic screen using *esg*$^{ts}$ to drive RNAi expression in precursors was conducted. Interestingly, expression of two independent RNAi lines VDRC12574 (*Figure 2*) and NIG12052 R-1 (*Figure 2—figure supplement 1*), both target and abolish *lola* expression (*Figure 2—figure supplement 1A*), resulted in an increased number of GFP$^+$ and p-H3$^+$ cells in the posterior midgut (*Figure 2A–C* and *Figure 2—figure supplement 1B–D*). Moreover, the p-H3$^+$ cells were dually labeled with Dl, suggesting that these cells are proliferative ISCs (*Figure 2A–C*). These results indicate that reducing *lola* expression in gut precursor cells promotes ISC proliferation. In addition, MARCM *lola*$^{5D2}$ (a hypomorphic allele) mutant clones were bigger in size compared to the control (*Figure 2D–E'''*), indicating a growth advantage due to ISC hyperproliferation when *lola* expression is reduced. As shown in *Figure 2E–E'''*, these *lola*$^{5D2}$ mutant clones contained Pdm1$^+$ ECs and Pros$^+$ ee cells, suggesting that Lola activity is not required for ISC differentiation.

In addition to the cell-autonomous role of Lola in precursor cells, we assess the possibility that Lola functions non-cell-autonomously by expressing *lola* RNAi in ECs. *MyoIA*$^{ts}$ is a driver combining *tub-Gal80*$^{ts}$ and *Myo IA-Gal4*, an enhancer trap inserted in the gut specific brush border *myosin IA* gene (*Morgan et al., 1994*). *Myo IA*$^{ts}$-driven GFP expression (MyoIA-GFP) labels ECs (*Figure 2F–F''*). Compared with the control groups, expressing either *lola* RNAi in ECs dramatically promoted ISC proliferation, as indicated by the increased Dl$^+$ and p-H3$^+$ cell number (*Figure 2F–H* and *Figure 2—figure supplement 1E–G*). Moreover, reducing *lola* expression in ECs resulted in gut hypertrophy associated with multi-layer of intestinal cells (*Figure 2I–J''*), possibly a consequence of hyperproliferation. Based on the findings that ECs interact with ISCs and regulate their proliferation (*Gervais and Bardin, 2017*), these results suggest a non-cell-autonomous role of Lola in regulating ISC proliferation during midgut homeostasis. Importantly, reducing *lola* expression in ECs did not induce non-specific apoptosis in the posterior midgut (*Figure 2—figure supplement 1H–I''*), indicating that *lola* RNAi promotes ISC proliferation via a specific signaling pathway. Altogether, Lola is required in both precursor cells and ECs for ISC proliferation and midgut homeostasis.

## Loss of Lola activates EGFR, JAK-STAT, and microRNA *bantam* signaling

Due to functional similarities between Lola and Wts in mediating midgut homeostasis, we next sought to determine whether Lola, like Wts, functions as a component in Hippo signaling. It has been shown that inactivation of Hippo signaling activates EGFR, JAK-STAT, and microRNA *bantam* pathways to stimulate ISC proliferation (*Karpowicz et al., 2010*; *Ren et al., 2010*; *Shaw et al.*,

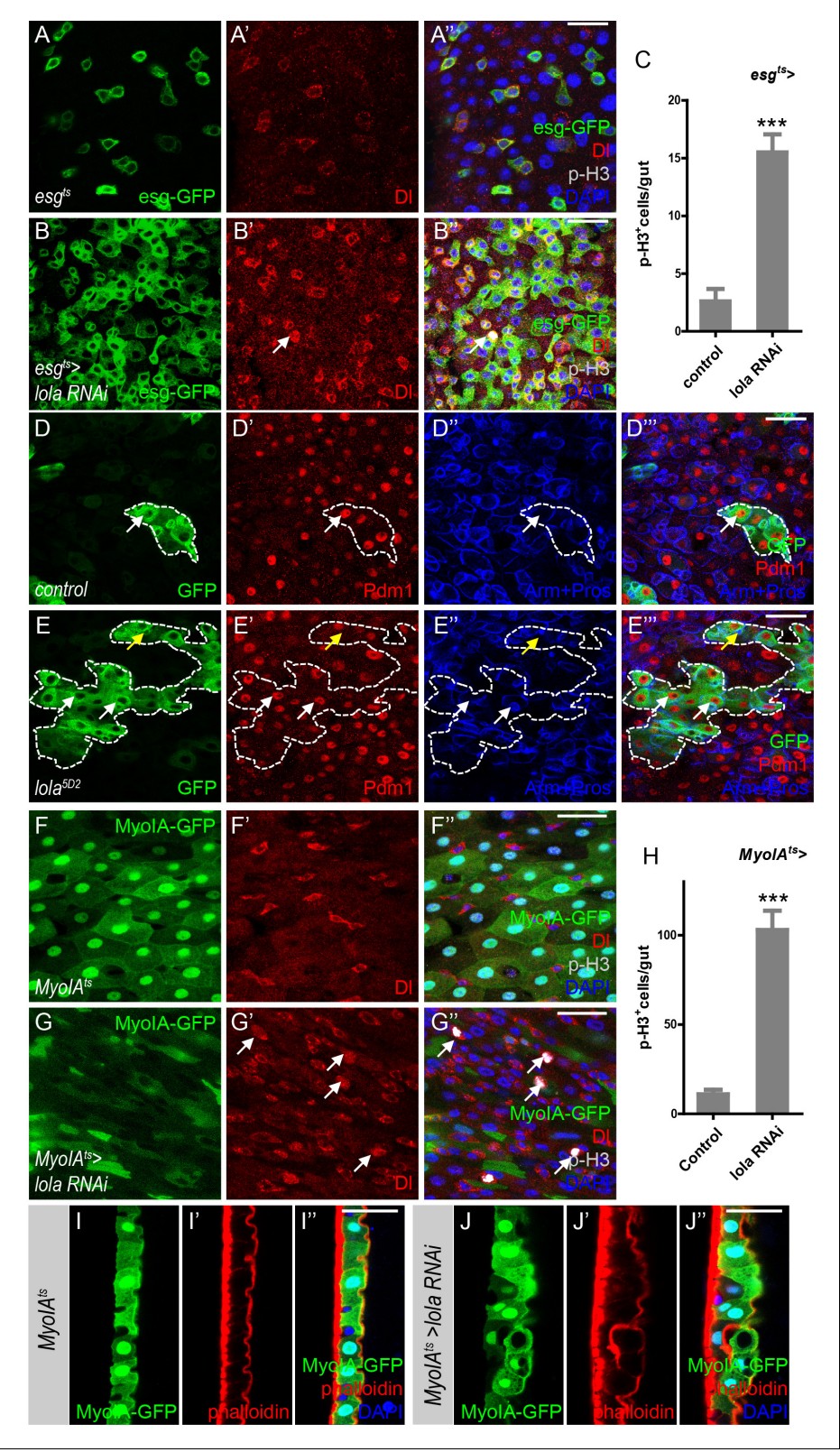

**Figure 2.** Lola is required for ISC proliferation and midgut homeostasis. (**A–B''**) Representative images of *Drosophila* adult midguts of *esg*<sup>ts</sup> (**A–A'**) and *esg*<sup>ts</sup> >*lola* RNAi (**B–B''**). Midguts were dissected and immunostained with antibodies against Dl (red), p-H3 (gray), and DAPI (nuclei, blue). ISCs and EBs are marked with esg-GFP (green). White arrows indicate proliferative ISCs marked by Dl and p-H3 (**B' and B''**). Merged images are shown in *Figure 2 continued on next page*

*Figure 2 continued*

A'' and B'' (green, red, gray, and blue). (**C**) Quantifications of the p-H3$^+$ cell number in adult midguts with the indicated genotypes in A and B (n = 9, 10). Note a significant increase in the p-H3$^+$ cell number when *lola* RNAi is expressed (***p<0.001). (**D–E'''**) Representative images of *Drosophila* adult midguts containing GFP positive MARCM clones of control (**D–D'''**) and the hypomorphic allele *lola*$^{5D2}$ (**E–E'''**). Midguts were dissected 4 days after clone induction and immunostained with antibodies against Arm and Pros (blue) and Pdm1 (red). Areas enclosed by the dashed lines indicate respective clone size. White arrows indicate Pdm1$^+$ ECs, and yellow arrows indicate Pros$^+$ ee cells. Merged images are shown in D''' and E'' (green, red, and blue). (**F–G''**) Representative images of *Drosophila* adult midguts of *MyoIA-Gal4; tubGal80*$^{ts}$ (*MyoIA*$^{ts}$) (**F–F''**) and *MyoIA*$^{ts}$ > *lola* RNAi (**G–G''**). Midguts were dissected and immunostained with antibodies against Dl (red), p-H3 (gray), and DAPI (blue). ECs are marked with MyoIA-GFP (green). White arrows indicate proliferative ISCs marked by Dl and p-H3 (**G'** and **G''**). Merged images are shown in F'' and G'' (green, red, gray, and blue). (**H**) Quantifications of the p-H3$^+$ cell number in adult midguts from the indicated genotypes in F and G (n = 10, 10). Note a significant increase in the p-H3$^+$ cell number when *lola* RNAi is expressed (***p<0.001). (**I–J''**) Representative images of cross section of *Drosophila* adult midguts from *MyoIA*$^{ts}$ (**I–I''**) and *MyoIA*$^{ts}$ > *lola* RNAi (**J–J''**) immunostained with antibodies against phalloidin (red) and DAPI (blue). Merged images are shown in I'' and J'' (green, red, and blue). Scale bars: 30 μm in A'', B'', D''', E'''; 25 μm in F'', G'', I'', J''. Data are shown as mean ± SEM. ***p<0.001 by Student's T-test. Confocal images were taken from the basal layer of the midgut where ISCs can be clearly visualized except in I-J''. Single layer image is shown.

The online version of this article includes the following source data and figure supplement(s) for figure 2:

**Source data 1.** Source data for *Figure 2C*, *Figure 2H*, *Figure 2—figure supplement 1D*, and *Figure 2—figure supplement 1G*.

**Figure supplement 1.** Lola is required for ISC proliferation and midgut homeostasis (A)Relative *lola* mRNA levels were analyzed by real-time PCR.

*2010*; *Staley and Irvine, 2010*). Interestingly, Lola depletion in either precursors or ECs caused a significant increase in the levels of dpERK, the dephosphorylated active form of MAPK (*Gabay et al., 1997*), and the mRNA levels of EGFR ligands *Spitz* (*Spi*), *Vein* (*Vn*), and *Krn*, all indicating EGFR activation (*Figure 3A–E*). Moreover, Lola depletion in either precursors or ECs activates JAK-STAT signaling as indicated by the elevated expressions of JAK-STAT ligand *Upd3* and *Stat-GFP*, a multimerized *Stat92E* reporter driving the expression of a destabilized GFP to monitor JAK-STAT activity (*Bach et al., 2007*) (*Figure 3F–M*). Consistently, mRNA levels of *Upds* (*Upd*, *Upd2*, and *Upd3*), and the JAK-STAT target *Socs36E* were elevated dramatically upon Lola depletion (*Figure 3N*). Furthermore, expression levels of the microRNA *bantam*, a canonical downstream target of Hippo signaling essential for cell-autonomous ISC proliferation (*Huang et al., 2014*), were also elevated upon Lola depletion (*Figure 3O–P'''*). These results demonstrate that loss of Lola activates EGFR, JAK-STAT, and microRNA *bantam* signaling, suggesting that Lola regulates ISC proliferation likely via mechanisms involving Hippo signaling.

## Lola regulates midgut homeostasis independently of Yki-Sd

We next investigate if Lola function depends on the core Yki-Sd complex. A driver of ubiquitous *Actin-Gal4* combined with *tub-Gal80*$^{ts}$ (*Actin*$^{ts}$) was used to manipulate *lola* expression ubiquitously in a temporal manner. Real-time PCR analysis of total RNAs from *Actin*$^{ts}$ > *lola* RNAi guts revealed no significant difference in *yki* mRNA levels between control and RNAi samples, suggesting that Lola does not affect *yki* transcription (*Figure 4A*). No Yki protein accumulation was detected in *lola*$^{5D2}$ mutant gut clones, suggesting that Lola does not affect Yki protein levels either (*Figure 4B–C'*). Genetic interaction analysis indicated that co-expression of either *yki* or *sd* RNAi with *lola* RNAi in ISCs did not suppress the increase in p-H3$^+$ cell number induced by *lola* RNAi alone (*Figure 4D* and *Figure 4—figure supplement 1A–F'*). Furthermore, MARCM clones of double mutant *lola*$^{5D2}$ and *yki*$^{B5}$ exhibited no significant difference in clone size compared to *lola*$^{5D2}$ (*Figure 4E–H'*). MARCM *sd*$^{ΔB1}$ mutant clones expressing *lola* RNAi were similarly larger in size, suggesting that Sd depletion does not suppress *lola* RNAi-induced ISC proliferation (*Figure 4I–L'*). Taken together, these results indicate that Lola restricts ISC proliferation independently of the Yki-Sd transcriptional complex.

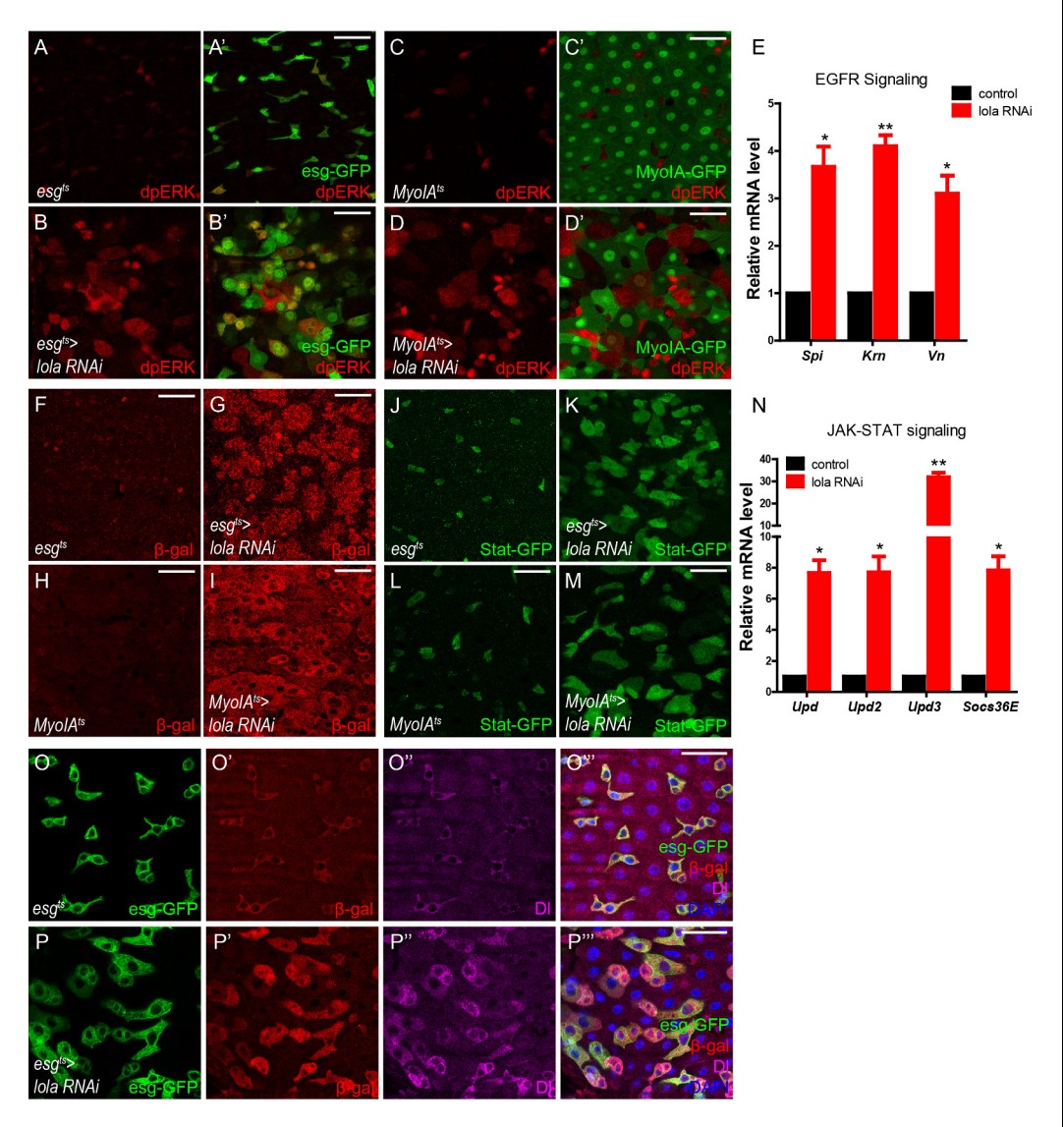

**Figure 3.** Loss of Lola activates EGFR, JAK-STAT, and microRNA *bantam* signaling. (**A–B'**) Representative images of *Drosophila* adult midguts of *esg*^ts^ (**A–A'**) and *esg*^ts^ >*lola* RNAi (**B–B'**) immunostained with dpERK (red). ISCs and EBs are marked with esg-GFP (green). Merged images are shown in A' and B' (green and red). Note the increased dpERK signal in *lola* RNAi guts compared with control. (**C–D'**) Representative images of *Drosophila* adult midguts of *MyoIA*^ts^ (**C–C'**) and *MyoIA*^ts^ > *lola* RNAi (**D–D'**) immunostained with dpERK (red). ECs are marked with MyoIA -GFP (green). Merged images are shown in C' and D' (green and red). Note the increased dpERK signal in *lola* RNAi guts compared with control. (**E**) Relative mRNA levels of EGFR ligands *Spi*, *Krn*, and *Vn* were analyzed by real-time PCR. Total RNAs were collected from whole midguts of the indicated genotypes: *MyoIA*^ts^ (black bars) and *MyoIA*^ts^ > *lola* RNAi (red bars). Note a significant increase in the mRNA levels when *lola* RNAi is expressed (*p<0.05 and **p<0.01). (**F–I**) Representative images of *Drosophila* adult midguts of *esg-Gal4; tubGal80*^ts^*/Upd3 LacZ* (*esg*^ts^*/Upd3 LacZ*) (**F**), *esg*^ts^*/Upd3 LacZ>lola* RNAi (**G**), *MyoIA-Gal4; tubGal80*^ts^*/Upd3 LacZ* (*MyoIA*^ts^*/Upd3 LacZ*) (**F**), and *MyoIA*^ts^*/Upd3 LacZ>lola* RNAi (**G**) immunostained with β-gal (red). Note an increase in *Upd3-LacZ* levels marked by β-gal when *lola* RNAi is expressed in either precursors or ECs. (**J–M**) Representative images of *Drosophila* adult midguts of *esg-Gal4/Stat GFP; tubGal80*^ts^ (*esg*^ts^*/Stat* GFP) (**J**), *esg*^ts^*/Stat GFP>lola* RNAi (**K**), *MyoIA-Gal4/Stat GFP; tubGal80*^ts^ (*MyoIA*^ts^*/Stat* GFP) (**L**), and *MyoIA*^ts^*/Stat GFP>lola* RNAi (**M**) immunostained with GFP (green). Note the increased *Stat-GFP* signal when *lola* RNAi is expressed in either precursors or ECs. (**N**) Relative mRNA levels of JAK-STAT ligands *Upd*, *Upd2*, *Upd3*, and EGFR downstream gene target *Socs36E* were analyzed by real-time PCR. Total RNAs were collected from whole midguts of the indicated genotypes: *MyoIA*^ts^ (black bars) and *MyoIA*^ts^ > *lola* RNAi (red bars). Note a significant increase in the mRNA levels when *lola* RNAi is expressed (*p<0.05 and **p<0.01). (**O–P'''**) Representative images of *Drosophila* adult midguts of *esg*^ts^*/bantam-LacZ* (**O–O'''**), and *esg*^ts^*/bantam-LacZ* >*lola* RNAi (**P–P'''**) immunostained with β-gal (red), Dl (magenta), and DAPI (nuclei, blue). ISCs and EBs are marked with esg-GFP (green). Merged images are shown in O''', and P''' (green, red, magenta, and blue). Note the increase in *bantam-LacZ* levels marked by β-gal when *lola* RNAi is expressed. Scale bars: 30 μm. Data are shown as mean ± SEM. *p<0.05 and **p<0.01 by Student's T-test. At least 10 midguts were dissected for each genotype. Confocal images were taken from the basal layer of the midgut where ISCs can be clearly visualized. Single layer image is shown.

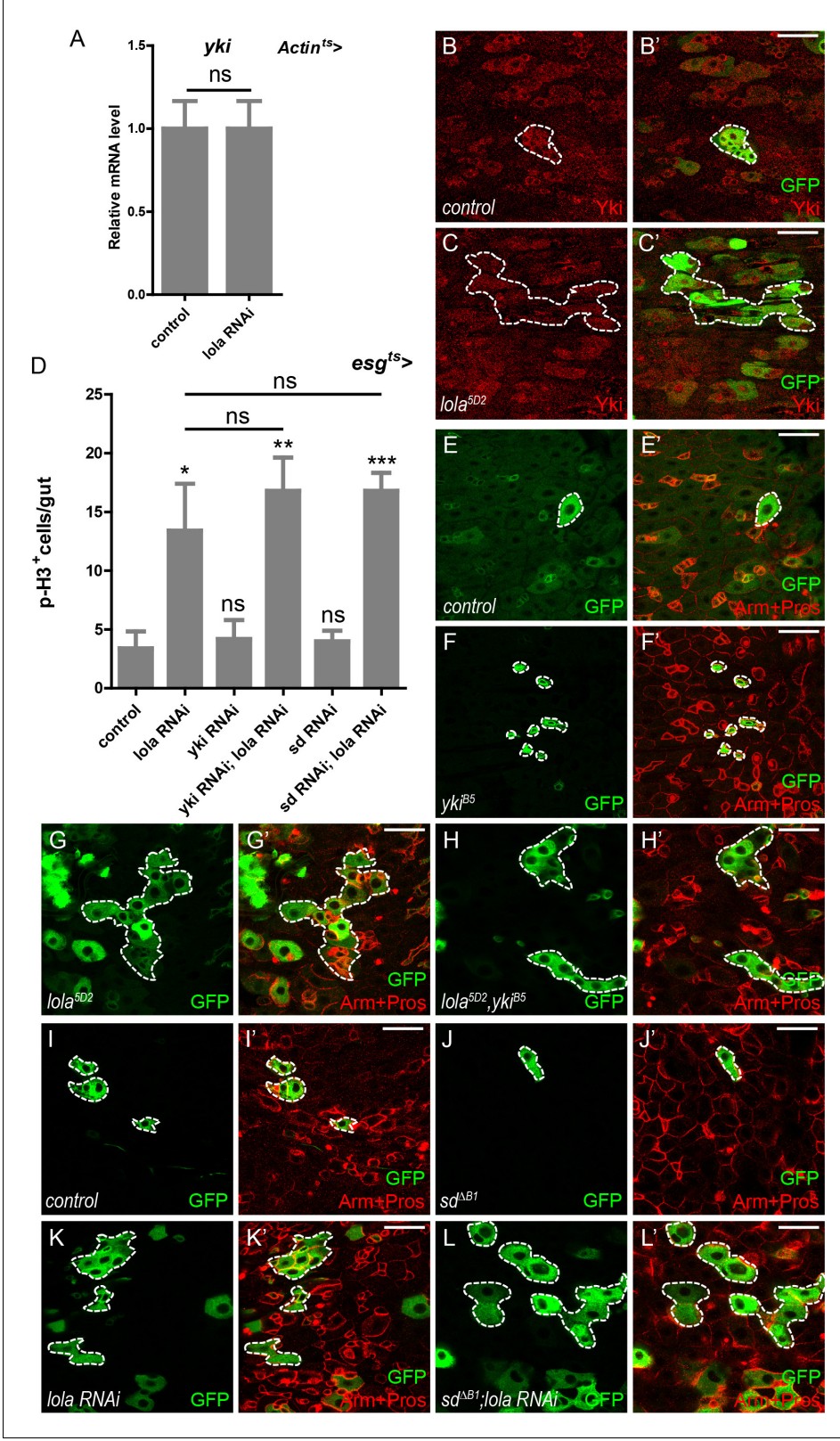

**Figure 4.** Lola regulates midgut homeostasis independently of Yki-Sd. (**A**) Relative *yki* mRNA levels were analyzed by real-time PCR. Total RNAs were collected from whole midguts of *Actin-Gal4; tub-Gal80ts* (*Actints*) and *Actints > lola* RNAi. Note that *yki* mRNA levels remain similar to the control when *lola* RNAi is expressed (ns p>0.05). (**B–C′**) Representative images of *Drosophila* adult midguts containing GFP positive MARCM clones of

*Figure 4 continued*

control (**B–B'**) and the hypomorphic allele *lola*$^{5D2}$ clones (**C–C'**). Midguts were dissected 2 days after clone induction and immunostained with antibodies against Yki (red). Areas enclosed by the dashed lines indicate respective clone size. Merged images are shown in B' and C' (green and red). Note that Yki protein levels are not affected in *lola*$^{5D2}$ mutant clones. (**D**) Quantifications of the p-H3$^+$ cell number in adult midguts from the indicated genotypes in *Figure 4—figure supplement 1A–F* (n = 9, 10, 10, 11, 11, 9). Note a significant increase in the p-H3$^+$ cell number when *lola* RNAi is expressed (*p<0.05). Co-expression of *yki* or *sd* RNAi does not suppress the increase (ns p>0.05). (**E–H'**) Representative images of *Drosophila* adult midguts containing GFP positive MARCM clones of control (**E–E'**), *yki*$^{B5}$ (**F–F'**), the hypomorphic allele *lola*$^{5D2}$ (**G–G'**), *yki*$^{B5}$ and *lola*$^{5D2}$ double mutants (**H–H'**). Midguts were dissected 2 days after clone induction and immunostained with antibodies against Arm and Pros (red). Areas enclosed by the dashed lines indicate respective clone size. Merged images are shown in E', F', G', and H' (green and red). Note that MARCM clones of *yki*$^{B5}$ and *lola*$^{5D2}$ exhibit similar size to *lola*$^{5D2}$. (**I–L'**) Representative images of *Drosophila* adult midgut containing GFP positive MARCM clones of control (**I–I'**), *sd*$^{ΔB1}$ (**J–J'**), *lola* RNAi (**K–K'**), and *sd*$^{ΔB1}$ in the presence of *lola* RNAi (**L–L'**). Midguts were dissected 2 days after clone induction and immunostained with antibodies against Arm and Pros (red). Areas enclosed by the dashed lines indicate respective clone size. Merged images are shown in I', J', K', and L' (green and red). Note that MARCM clones of *sd*$^{ΔB1}$ expressing *lola* RNAi exhibit similar size to clones expressing *lola* RNAi alone. Scale bars: 30 μm. Data are shown as mean ± SEM. ns p>0.05, *p<0.05, **p<0.01, and ***p<0.001 by Student's T-test. At least 10 midguts were dissected for each genotype. Confocal images were taken from the basal layer of the midgut where ISCs can be clearly visualized. Single layer image is shown.

The online version of this article includes the following source data and figure supplement(s) for figure 4:

**Source data 1.** Source data for *Figure 4D*.
**Figure supplement 1.** Lola regulates midgut homeostasis independently of Yki-Sd.

## Lola restricts ISC proliferation by suppressing *Dref* and *SkpA* expression levels

To further elucidate the mechanism of Lola regulating ISC proliferation and midgut homeostasis, we turn to its function as a DNA binding protein and transcriptional suppressor (*Southall et al., 2014*). Chromatin immunoprecipitation assay followed by high-throughput sequencing (ChIP-seq) was performed to identify genes suppressed directly by Lola. Thousands of Lola-associated chromatin binding sites were identified in cultured S2 cells. Analysis of the Lola binding profiles revealed that Lola mainly binds to the regions around transcription start sites (TSS) and promoters. Among the top 500 defined scored peaks (Supplementary file-Table 1), genes related to stem cell mitosis and differentiation were manually selected for further analysis. Results from ChIP assay and real-time PCR showed dramatic enrichment of Lola in promoter regions of *Oli*, *ec*, *Dref*, and 3'UTR regions of *pdm3* and S*kpA*. No significant binding (only a basal level of binding) was detected between Lola and the non-binding region (*genebody, gb*) (*Figure 5A*). As a negative control, Lola knockdown by dsRNA significantly reduced the enrichment (*Figure 5A*). To validate these results in vivo, total RNAs from *Actin*$^{ts}$ > *lola* RNAi guts were collected and analyzed. Similarly, mRNA levels of *pdm3*, *ec*, *Dref*, and *SkpA* were upregulated when reducing *lola* expression, suggesting that Lola suppresses the expression of these genes (*Figure 5B*). To further confirm that Lola represses the transcription of these genes, Lola binding regions (1: long and 2: short) of *Dref* or *SkpA*, two target genes among all others consistently affected by Lola, were cloned into upstream or downstream of the *luciferase* gene and the reporter activity was assayed. As shown in *Figure 5C*, activities of these reporters were dramatically inhibited by Lola but not LolaΔZF12, a truncated form of Lola that does not bind to DNA. Activities of *3xSd luc*, a dual reporter reflecting Sd-Yki transcriptional activity (*Zhang et al., 2008*), was not affected by Lola, demonstrating the specificity of Lola repression. In addition, Dref protein levels were significantly elevated in *lola*$^{5D2}$ mutant clones (*Figure 4D–F*). In summary, these results suggest that Lola regulates the expression of *Dref* and *SkpA*, and represses their transcription.

To verify whether these genes are responsible for the ISC hyperproliferation induced by *lola* RNAi, ISC proliferation was monitored in midguts co-expressing RNAi targeting *pdm3*, *ec*, *Dref* or *SkpA* and *lola*. As expected, knock down of either *Dref* or *SkpA* significantly rescued the increase in GFP$^+$ and p-H3$^+$ cell number induced by *lola* RNAi (*Figure 5G* and *Figure 5—figure supplement 1A–H'*), indicating that Lola restricts ISC proliferation via regulating *Dref* or *SkpA* expression levels.

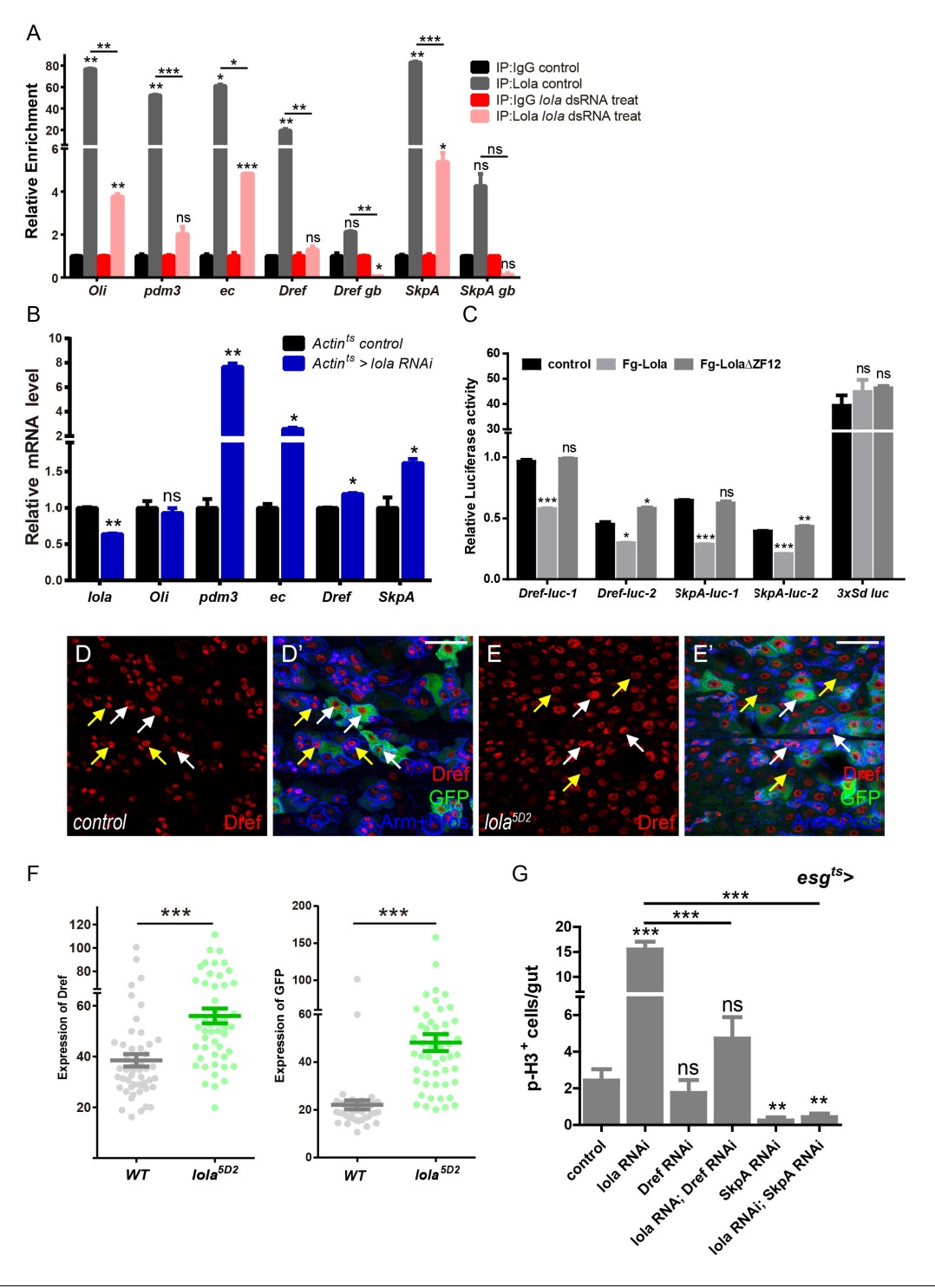

**Figure 5.** Lola restricts ISC proliferation by suppressing *Dref* and *SkpA* expression levels. (**A**) Relative enrichment of Lola on binding and non-binding (*genebody, gb*) regions compared to IgG in control (*Renilla* dsRNA treated, gray and black bars) and Lola knockdown cells (*lola* dsRNA treated, pink and red bars) were analyzed by ChIP and real-time PCR. Note a significant increase of Lola enrichment in regions near *pdm3*, *ec*, *Dref* and *SkpA* (ns p>0.05, *p<0.05, **p<0.01, and ***p<0.001). (**B**) Relative mRNA levels of the indicated genes analyzed by real-time PCR. Total RNAs were collected from whole midguts of the indicated genotypes: *Actin*^*ts* (black bars) and *Actin*^*ts* > *lola* RNAi (blue bars). Note a significant increase in the mRNA levels of *pdm3*, *ec*, *Dref* and *SkpA* when *lola* RNAi is expressed (ns p>0.05, *p<0.05, and **p<0.01). (**C**) Relative luciferase activity of the luciferase reporter plasmids

*Figure 5 continued on next page*

*Figure 5 continued*

carrying Lola-binding-region on *Dref* or *SkpA* in S2 cells transfected with the indicated constructs. The luciferase reporter plasmid with 3xSd-binding sites (*3xSd luc*) was used as control. (ns p>0.05, *p<0.05, **p<0.01, and ***p<0.001) (D–E') Representative images of *Drosophila* adult midguts containing GFP positive MARCM clones of control (D–D') and the hypomorphic allele *lola$^{5D2}$* (E–E'). Midguts were dissected 4 days after clone induction and immunostained with antibodies against Dref (red) and Arm and Pros (blue). White arrows indicate GFP positive clonal cells and yellow arrows indicate adjacent control cells. Merged images are shown in C' and D' (green, red, and blue). Note an increase on Dref protein levels in the GFP positive *lola$^{5D2}$* mutant clones (white arrows in E and E'). (F) Quantification of Dref expression in E. (G) Quantifications of the p-H3$^+$ cell number in adult midguts with the indicated genotypes in *Figure 5—figure supplement 1C–H* (n = 7, 8, 8, 7, 8, 7). Note a significant increase in the p-H3$^+$ cell number when *lola* RNAi is expressed (***p<0.001). Co-expression of *Dref* or *SkpA* RNAi and *lola* RNAi suppresses the increase (***p<0.001). Scale bars: 30 μm. Data are shown as mean ± SEM. ns p>0.05, *p<0.05, **p<0.01, and ***p<0.001 by Student's T-test. At least 10 midguts were dissected for each genotype. Confocal images were taken from the basal layer of the midgut where ISCs can be clearly visualized. Single layer image is shown.

The online version of this article includes the following source data and figure supplement(s) for figure 5:

**Source data 1.** Source data for *Figure 5F* and *Figure 5G*.
**Figure supplement 1.** Lola restricts ISC proliferation by suppressing *Dref* and *SkpA* expression levels.

On the other hand, knock down of either *Pdm3* or *ec* did not rescue the increased proliferative ISC number induced by *lola* RNAi (data not shown), indicating that these target genes might be involved in other parts of Lola function. Based on these results, we conclude that Lola functions as a transcription factor to suppress the expression of mitosis-related genes *Dref* and *SkpA*, hence controlling ISC proliferation and midgut homeostasis.

## Wts interacts with lola and affects lola turnover in vitro

Given that both Lola and Wts regulate ISC proliferation, and Lola deactivates similar downstream pathways as Hippo signaling, biochemical approaches were first taken to study the relationship between Hippo signaling components and Lola in cultured S2 cells. Hippo signaling components such as Hpo, Wts, Yki, Sb, or SdBP engineered with different N-terminal epitope tags for identification (Fg-Hpo, Myc-Wts, Fg-Yki, HA-Sd, or Myc-SbBP) were co-expressed with Lola tagged with a N-terminal Flag epitope (Fg-Lola). Interestingly, Fg-Lola protein levels were significantly increased only in the presence of Myc-Wts (*Figure 6A*). In addition, Fg-Lola was increasingly stabilized when adding increasing amount of Myc-Wts, indicating that Wts regulates Lola protein levels in a dosage-dependent manner (*Figure 6B*). Furthermore, Fg-Lola protein half-life in the presence or absence of Myc-Wts was monitored using the nascent protein synthesis inhibitor cycloheximide (CHX). As shown in *Figure 6C–D*, Fg-Lola proteins exhibited a rapid turnover rate with a half-life of approximately 2 hr, while Myc-Wts prolonged Fg-Lola protein half-life to about 6 hr. To investigate how Lola protein stability is regulated, S2 cells expressing Fg-Lola were treated with specific UPS inhibitors (MG132 or MG101) or lysosomal inhibitors (E64 or Leupeptin) for 3 hr followed by CHX treatment for additional 6 hr. As shown in *Figure 6—figure supplement 1A*, UPS inhibitors, but not lysosome inhibitors, protected Fg-Lola from degradation, indicating that UPS plays a major role in regulating Lola protein stability. In addition, Lola ubiquitination levels were reduced in the presence of Wts as shown by results from the in vivo ubiquitination assay (*Figure 6—figure supplement 1B*). These results demonstrate that Wts regulates Lola protein stability by protecting Lola from UPS-mediated degradation, hence the possibility of Hippo signaling regulating Lola function.

To further explore possible interaction between Wts and Lola, co-immunoprecipitation (Co-IP) assays in S2 cells expressing Myc-Wts and Fg-Lola were conducted. As shown in *Figure 6E*, Lola and Wts reciprocally interacted, validating a physical interaction between two proteins. Co-IP analysis of S2 cells expressing Myc-Wts using the anti-Lola antibody for IP pulled down Myc-Wts, further demonstrating that Wts interacts with endogenous Lola (*Figure 6F*). To map which Lola region binds to Wts, two truncated Lola protein variants, each engineered with a Flag tag, were constructed: the N-terminal Lola (amino acid 1 to 370) containing the BTB domain (Fg-LolaN) and the C-terminal Lola (amino acid 371 to 748) containing the two zinc finger motifs (ZF1 and ZF2, Fg-LolaC) (*Figure 6—figure supplement 2A*). Co-IP results indicated that Fg-LolaC, but not Fg-LolaN, co-

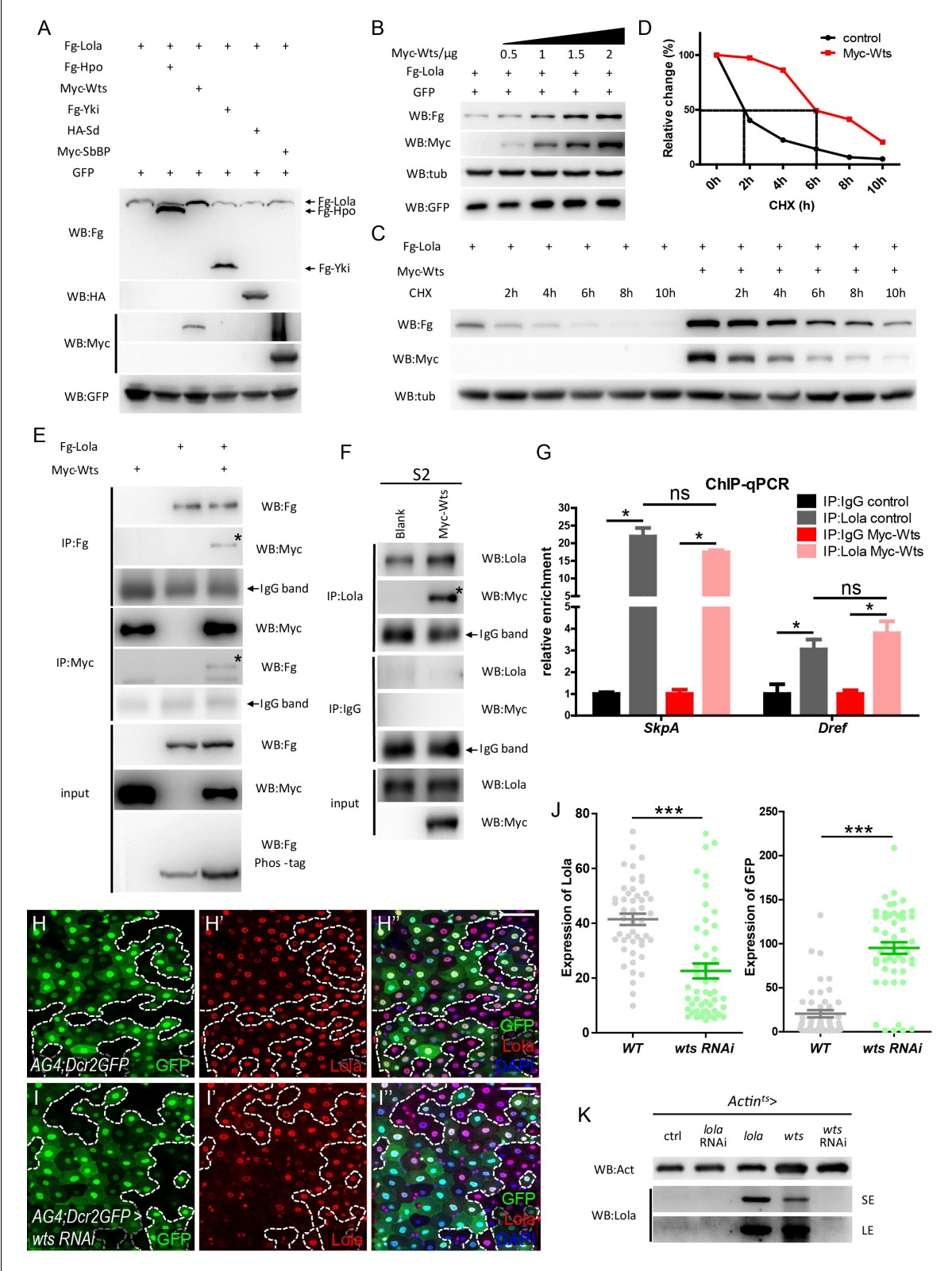

**Figure 6.** Wts interacts with Lola and affects Lola protein turnover. (**A**) S2 cells were co-transfected with constructs expressing different Hippo components (Fg-Hpo, Myc-Wts, Fg-Yki, HA-Sd, or Myc-SbBP) and Fg-Lola. Cell extracts were collected and analyzed by Western blot with the indicated antibodies. Note that Lola protein levels were specifically modulated by Wts. (**B**) Collected cell extracts from S2 cells co-transfected with plasmids expressing *Fg-Lola* (1 μg) and *Myc-Wts* (0.5, 1, 1.5, and 2 ug) were analyzed by Western blot with the indicated antibodies. Plasmids expressing *GFP*

Figure 6 continued on next page

Figure 6 continued

were also added as a control. Note that Lola is stabilized by Wts in a dosage-dependent manner. (C) S2 cells were transfected with plasmids expressing *Fg-Lola* with or without *Myc-Wts* and treated with CHX for the indicated hours. Cells were harvested and analyzed by Western blot with the indicated antibodies. Note that Lola half-life extends in the presence of Wts. (D) Quantifications of results normalized to Tubulin in C. Note that Lola half-life extends from 2 to 6 hr in the presence of Myc-Wts. (E) Co-IP analysis indicates that Fg-Lola and Myc-Wts co-immunoprecipitate with each other using antibodies against either Flag or Myc. Pull-down proteins are marked by the asterisks. (F) Co-IP analysis indicates that Myc-Wts interacts with endogenous Lola using antibodies against Lola or Myc. Pull-down proteins are marked by the asterisk. (G) Relative enrichment of Lola on binding regions of *SkpA* and *Dref* compared to IgG in control (gray and black bars) and cells expressing *Myc-Wts* (pink and red bars) were analyzed by ChIP and real-time PCR. Note that Wts does not affect the significant increase of Lola enrichment in these regions (ns p>0.05 and *p<0.05). (H–I''')
Representative images of *Drosophila* adult midguts containing GFP positive flip-out clones of *hsflp; act >CD2>Gal4; UAS-Dicer2, UAS-GFP* (AG4; Dcr2GFP) (H–H''') and *AG4; Dcr2GFP >wts* RNAi (I–I'''). Midguts were dissected 2 days after clone induction and immunostained with antibodies against Lola (red) and DAPI (nuclei, blue). Areas enclosed by the dashed lines indicate clone regions, respectively. Merged images are shown in H''' and I''' (green, red, and blue). Note a decrease in Lola protein levels in GFP positive flip-out clones expressing *wts* RNAi (white arrows I-I'''). (J) Quantification of Lola expression in I. (K) Western blot analysis of gut extracts collected from adult flies with the indicated genotypes (*Actin^ts > lola, wts, lola* RNAi, or *wts* RNAi). Note that Lola protein levels were increased when *lola* or *wts* is overexpressed. SE: short exposure. LE: long exposure. Scale bars: 30 μm. Data are shown as mean ± SEM. ns p>0.05, *p<0.05 by Student's T-test. At least 10 midguts were dissected for each genotype. Confocal images were taken from the basal layer of the midgut where ISCs can be clearly visualized. Single layer image is shown.

The online version of this article includes the following source data and figure supplement(s) for figure 6:

Source data 1. Source data for *Figure 6J*.
Figure supplement 1. Wts affected Lola UPS-mediated degradation.
Figure supplement 2. Wts interacts with C-terminal Lola independently of Wts kinase activity and Lola zinc finger domains.
Figure supplement 3. Wts affects Lola protein levels in vivo.

immunoprecipitated with Myc-Wts (*Figure 6—figure supplement 2B*), suggesting that Wts interacts with Lola C-terminus. Deletion of either ZF1 or ZF2 or both in Lola C-terminus did not affect binding between Wts and Lola, suggesting that Wts-Lola interaction does not require DNA binding (*Figure 6—figure supplement 2A and C*). Furthermore, ChIP-qPCR analysis revealed no detectable change of Lola enrichment on *Dref* or *SkpA* binding regions when co-expressing Wts in S2 cells (*Figure 6G*), indicating that Wts does not affect Lola-DNA binding. *In toto*, these results suggest that Wts interacts with Lola C-terminus independently of Lola binding to DNA.

Given that Wts is a Serine/Threonine protein kinase, we next sought to determine if Lola is a Wts substrate and phosphorylated by Wts. Surprisingly, Lola exhibited no shift in molecular weight when analyzed by the phos-tag gels (*Figure 6E*), suggesting that Wts interacts with Lola in a manner independent of phosphorylation. To further validate this conclusion, a construct expressing Wts KD (Wts$^{K743R}$, a kinase dead form of Wts) (*Lai et al., 2005*) was generated. Not only that Wts KD interacted with Lola by Co-IP (*Figure 6—figure supplement 2C*), Wts KD stabilized Lola similarly as the wild-type Wts (data not shown). These results suggest that Wts kinase activity is not essential for binding to Lola and does not affect Lola protein stability. On the other hand, previous studies have suggested that Wts interacts and phosphorylates Yki (*Goulev et al., 2008*; *Huang et al., 2005*; *Oh and Irvine, 2008*). In our hands, both total amount and the S168 phosphorylation level of Yki co-immunoprecipitated with Wts remained unaffected in the presence or absence of Fg-Lola, suggesting that Wts-Lola interaction does not affect Wts-Yki interaction and Wts-mediated Yki phosphorylation (*Figure 6—figure supplement 2D*). Taken together, these results demonstrate that Wts binds to Lola directly to moderate its turnover rate, a process independent of Wts kinase activity, Lola DNA binding, Wts-Yki interaction, and Wts-mediated Yki phosphorylation.

It is noteworthy to mention that both Wts and Lola proteins have been predicted to contain the nuclear export signal (NES) (http://www.cbs.dtu.dk/services/NetNES/) and the nuclear localization signal (NLS) (http://nls-mapper.iab.keio.ac.jp/cgi-bin/NLS_Mapper_form.cgi), indicating that Wts possibly localizes to the nucleus in addition to the cytoplasm, whereas Lola localizes to the cytoplasm in addition to the nucleus. Our results showed that expression of Wts fused with a C-terminal nuclear localization signal (NLS) (Wts$^{NLS}$) caused a remarkable increase in Lola protein levels, suggesting that Wts$^{NLS}$ stabilizes Lola in the nucleus (*Figure 6—figure supplement 1C–D*), whereas the expression of Wts fused with a C-terminal nuclear export signal (NES) (Wts$^{NES}$) only reduced the increased Lola protein levels partially (*Figure 6—figure supplement 1C–D*), suggesting the possibility of Lola translocating to cytoplasm where it is protected by Wts. These results indicate that Wts interacts with Lola both in the cytoplasm and nucleus.

## Wts regulates lola protein levels in vivo

In addition to our evidence that Wts interacts with and stabilizes Lola in S2 cells, endogenous Lola protein levels were examined using antibodies against Lola. Immunostainings showed that Lola is ubiquitously expressed in all cell types in the midgut and absent in the *lola*$^{5D2}$ MARCM clones (*Figure 6—figure supplement 3A–B'''*). Consistent with the cell culture data, a dramatic decrease in the anti-Lola staining was also detected in both flip-out clones expressing *wts* RNAi using the driver *hsflp; act >CD2>Gal4; UAS-Dicer2, UAS-GFP* (*AG4; Dcr2GFP > wts* RNAi) and *wts*$^{x1}$ MARCM clones in midguts (*Figure 6H–J* and *Figure 6—figure supplement 3C–D'''*). Furthermore, endogenous Lola protein levels were analyzed by Western blot using gut extracts that ubiquitously express *lola*, *lola* RNAi, *wts*, or *wts* RNAi. Similar to *lola* overexpression, *wts* overexpression increased Lola protein levels (*Figure 6K*). Collectively, these results indicate that Wts regulates Lola protein levels in vivo.

## Lola is essential for Wts-mediated ISC proliferation and midgut homeostasis

Based on the results that Wts modulates Lola protein stability and Lola is required for midgut homeostasis, genetic interaction between Wts and Lola in regulating midgut homeostasis was analyzed. Whereas reduced *yki* expression did not rescue the increase in the GFP$^+$ and p-H3$^+$ cell number induced by *esg*$^{ts}$ > *wts* RNAi (*Figure 1A–E*), *lola* overexpression dramatically reduced this increase (*Figure 7A* and *Figure 7—figure supplement 1A–D'*). Similar reduction was detected in midguts expressing *lola* and *wts* RNAi in ECs (*MyoIA*$^{ts}$ > *lola; wts* RNAi, *Figure 7—figure supplement 1E*). These results indicate that *lola* overexpression blocks the hyperproliferation induced by *wts* RNAi both cell-autonomously and non-cell-autonomously. To further validate our observation, expressing *lola* in GFP$^+$*wts*$^{x1}$ MARCM clones that exhibited the overgrowth phenotype dramatically reduced the clone size (*Figure 7B–F*). On the other hand, *wts* overexpression in *lola*$^{5D2}$ MARCM clones did not affect the increased clone size (*Figure 7—figure supplement 2A–D*). Taken together, these results suggest that Lola is essential for Wts function in midgut homeostasis.

Furthermore, *Dref* and *SkpA* mRNA levels were also elevated in *wts*$^{x1}$ clonal midguts (*Figure 7—figure supplement 1F*). Genetic interaction analysis between *wts* and *Dref* or *SkpA* showed that co-expression of *SkpA* or *Dref* RNAi in *wts* RNAi midguts dramatically reduced the increase in the GFP$^+$ and p-H3$^+$ cell number (*Figure 7A*, *Figure 7—figure supplement 1E and G–H'*), indicating that SkpA and Dref are both required for Wts function in midgut homeostasis. These results support our hypothesis that Wts stabilizes Lola such that *SkpA* and *Dref* expressions are transcriptionally suppressed, thereby restricting ISC proliferation.

Finally, RNA sequencing (RNA-seq) analysis using RNAs isolated from adult guts expressing *wts* RNAi, *lola* RNAi or *yki* using *Actin*$^{ts}$ identified a collection of 2116 differentially expressed genes (DEGs), using the R package DESeq2 (padj <0.05) (*Figure 7G*, *Supplementary file 2*) (*Love et al., 2014*). Based on the DEG heatmap and hierarchical cluster analysis, a high degree of similarity was detected between samples from *wts* and *lola* RNAi, suggesting strong functional correlation between Wts and Lola, but not Yki and Lola, in regulating midgut homeostasis. On the other hand, *lola* overexpression did not rescue the elevated GFP$^+$ and p-H3$^+$ cell number induced by *yki* overexpression (*Figure 7—figure supplement 3A–E*), reinforcing the notion that Lola restricts ISC proliferation in an Yki-independent manner. Taken together, these results suggest that while Wts regulates Lola and Yki in different means, Lola are functionally similar with Wts. Yki and Lola regulate the expression of different target genes, and direct different transcription programs in the process of Hippo-mediated ISC proliferation and midgut homeostasis (*Figure 7H*).

## Discussion

Mechanisms that regulate *Drosophila* midgut homeostasis and regeneration are complex and under a series of delicate controls. The Hippo pathway, which output is mainly transduced by the effector Yki-Sd complex, has been shown to be crucial for both midgut homeostasis and regeneration. In the present study, we identify a novel transcription factor Lola acting downstream of Hippo signaling to restrict ISC proliferation. The Hippo component Wts interacts with and stabilizes Lola in a Yki-independent manner. Lola then suppresses downstream *Dref* and *SkpA* expression levels to regulate ISC

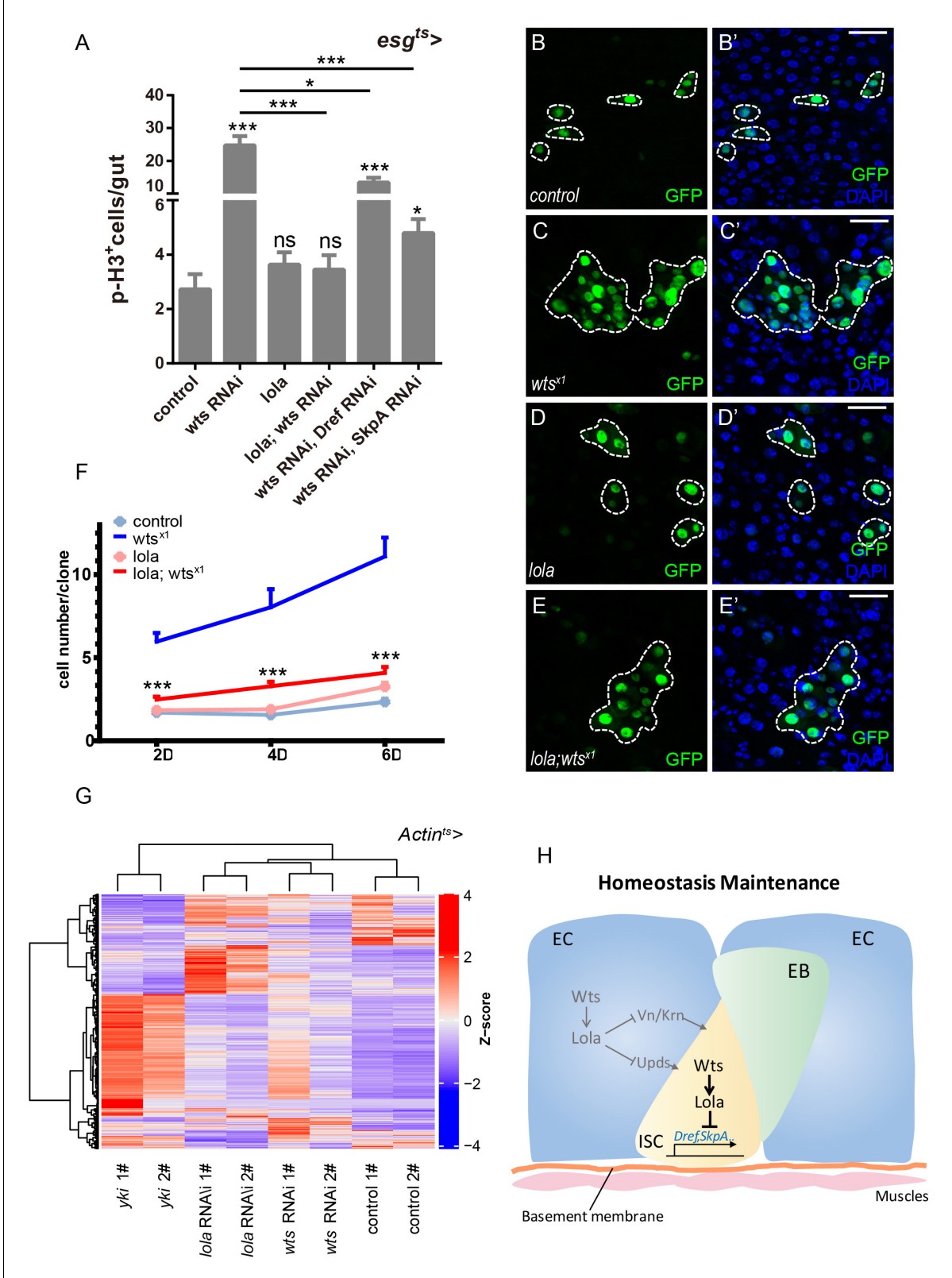

**Figure 7.** Lola is essential for Wts-mediated ISC proliferation and midgut homeostasis. (**A**) Quantifications of the p-H3$^+$ cell number in adult midguts with the indicated genotypes in *Figure 7—figure supplement 1A–F* (n = 18, 21, 19, 20, 18, 19). Note that co-expression of *lola*, *Dref* RNAi, or *SkpA* RNAi in the presence of *wts* RNAi suppresses the increase in the p-H3$^+$ cell number (ns p>0.05, *p<0.05, and ***p<0.001). (**B–E'**) Representative images of *Drosophila* adult midguts containing GFP positive MARCM clones of control (**B–B'**), *wts$^{x1}$* (**C–C'**), *lola* (**D–D'**), and *wts$^{x1}$* in the presence of *lola* (**E–E'**). *Figure 7 continued on next page*

*Figure 7 continued*

Midguts were dissected 4 days after clone induction and immunostained with antibodies against DAPI (nuclei, blue). Areas enclosed by the dashed lines indicate respective clone size. Merged images are shown in B', C', D', and E' (green and blue). Note a decrease in the $wts^{x1}$ clone size in the presence of *lola*. (F) Quantifications of the cell number per clone in adult midguts from the indicated genotypes in B-E at 2, 4, and 6 days after clone induction (***p<0.001). Average of 29–92 clones from 10 midguts for each genotype were quantified. (G) Heatmap of 2116 union DEGs in adult guts carrying genotypes: $Actin^{ts}$, $Actin^{ts} > yki$, $Actin^{ts} > lola$ RNAi, and $Actin^{ts} > wts$ RNAi. A decrease and an increase in expression is indicated by blue and red color, respectively. Note a high degree of similarity in the DEG expression profiles for *lola* and *wts* RNAi. (H) An illustrated model on *Drosophila* midgut homeostasis regulated by the Wts-Lola-Dref/SkpA signaling axis. Wts interacts with Lola and regulates its stability. In the absence of Lola, ISC undergoes hyperproliferation due to de-repression of *Dref* and *SkpA* expression levels. Yki and Lola regulate different target gene expression levels, thereby controlling midgut homeostasis via separate means. Scale bars: 30 μm. Data are shown as mean ± SEM. ns p>0.05, *p<0.05, and ***p<0.001 by Student's T-test. At least 10 midguts were dissected for each genotype. Confocal images were taken from the basal layer of the midgut where ISCs can be clearly visualized. Single layer image is shown.

The online version of this article includes the following source data and figure supplement(s) for figure 7:

**Source data 1.** Source data for *Figure 7A*, *Figure 7F*, *Figure 7—figure supplement 1E*, *Figure 7—figure supplement 2D*, and *Figure 7—figure supplement 3E*.

**Figure supplement 1.** Lola is essential for Wts-mediated midgut homeostasis.
**Figure supplement 2.** Lola is essential for Wts-mediated midgut homeostasis.
**Figure supplement 3.** Lola and Yki regulate the midgut homeostasis in different means.

proliferation. Our results suggest that Lola is an effector of a non-canonical Hippo signaling pathway essential for ISC proliferation and midgut homeostasis (*Figure 7H*).

## Lola is a new player in Hippo-mediated ISC proliferation and midgut homeostasis

Our present findings suggest that Lola regulates ISC proliferation and midgut homeostasis. Reduced *lola* expression in both ISCs and ECs promotes ISC proliferation, indicating that Lola functions both cell-autonomously and non-cell-autonomously. These precursors remain correctly differentiated when Lola is absent, suggesting that Lola only affects ISC proliferation, but not differentiation. Similarly, inactivation of Hippo signaling also causes enhanced ISC proliferation, indicating the possibility that Lola functions downstream of Hippo. Consistent to the characterized phenotype for Hippo signaling inactivation, downregulating *lola* expression in ECs activates EGFR and JAK-STAT signaling, and Lola regulates Hippo downstream *bantam* expression levels in ISCs. Taken together, these results suggest that Lola regulates ISC proliferation and homeostasis likely via mechanisms involving Hippo signaling.

Our biochemical results strongly support that Lola is a Hippo component. Not only that Lola interacts with and is stabilized by Wts, our in vivo analysis of ISC proliferation also indicates that *lola* overexpression rescues the enhanced ISC proliferation induced upon Wts depletion, a phenotype not affected by Yki. These findings indicate that Lola functions downstream of Hippo signaling via interaction with Wts. In addition, four lines of evidence suggest that Lola functions independently of the Yki-Sd complex: first, *yki* mRNA and protein levels are not affected by Lola, suggesting that *yki* is not a Lola transcriptional target. Genetic interaction analysis also shows that neither Yki nor Sd suppresses the enhanced ISC proliferation induced by *lola* RNAi. Next, RNA-seq transcriptional profiles of samples expressing *lola* RNAi strongly correlate with the ones expressing *wts* RNAi, but not the ones expressing *yki*, indicating that Yki and Lola work downstream of Wts to mediate different outputs in terms of regulating ISC proliferation. Furthermore, Lola-Wts interaction does not affect Wts-Yki binding and Wts-mediated Yki phosphorylation, suggesting Wts regulates Yki and Lola via separate means. Finally, co-expression of *lola* and *yki* does not rescue the enhanced ISC proliferation induced by *yki* overexpression, indicating that Lola and Yki work independently from each other. These results demonstrate that Lola is a novel effector of Hippo signaling in regulating ISC proliferation and midgut homeostasis; Lola functions by interacting with Wts and in a manner independent of the Yki-Sd complex.

## Mechanisms of Lola-mediated ISC proliferation and midgut homeostasis

As a transcriptional suppressor, it is intriguing to speculate that Lola regulates downstream gene expression by suppressing their transcription levels, hence affecting ISC proliferation. Using ChIP-

seq analysis and luciferase reporter assays, two distinct gene targets of Lola were identified: *Dref* and *SkpA*. Lola enrichment was detected on the *Dref* promoter and the 3'UTR region of *SkpA*, respectively, indicating that the repression modes of Lola on *Dref* and *SkpA* are different. Transcription of these two genes are also directly repressed by *lola*. Downregulation of *Dref* or *SkpA* expression in gut precursors expressing either *wts* or *lola* RNAi rescues the enhanced ISC proliferation, providing the genetic evidence that Lola regulates ISC proliferation vis suppressing *Dref* and *SkpA* expression. Wts-mediated ISC proliferation also requires Lola and its downstream transcription targets *Dref* and *SkpA*. Moreover, co-expression of *Dref* RNAi and *yki* does not rescue the enhanced ISC proliferation induced by *yki* overexpression, reconfirming the Yki-independent mechanism of Lola repressing Dref/SkpA signaling in regulating ISC proliferation and midgut homeostasis (data not shown).

## Wts regulates lola stability

Due to its important role in ISC proliferation and midgut homeostasis, Lola protein levels are expected to be under precise and delicate controls. Interestingly, Wts physically interacts with Lola and Lola protein levels are regulated by Wts both in vitro and in vivo. It has been proposed that Wts and Mats colocalize in the cytoplasm or at the cell cortex, thereby restricting Yki in the cytoplasm (*Dong et al., 2007*; *Oh and Irvine, 2008*; *Sun et al., 2015*). Exceptions have been reported, however, that activated LATS1/2 (Wts in mammals) is accumulated in the nucleus, where it phosphorylates YAP (Yki in mammals) (*Li et al., 2014*). Based on these observations, it is feasible to speculate that Wts translocates to the nucleus and interacts with Lola, therefore regulating Lola protein stability. Our cell culture data further indicate a possibility of Lola translocation into the cytoplasm, and Wts interacts with Lola both in cytoplasm and nucleus. Despite the possibility of Wts translocation into the nucleus (or Lola translocation into the cytoplasm), Wts is unlikely to regulate Lola stability via phosphorylation, an event happened for Yki in the cytoplasm based on our results that Wts kinase activity is not required for Wts-Lola interaction. Furthermore, Wts-Lola interaction is independent of Wts-Yki interaction and Wts-mediated Yki phosphorylation, demonstrating that Wts controls Yki and Lola activity via separate and distinct mechanisms.

Our biochemical results provide further insights on possible mechanisms of how Wts controls Lola stability. Wts interacts with the C-terminal Lola, where the DNA binding zinc finger motifs reside. Deletion of zinc finger motifs does not affect Wts-Lola binding, nor Wts affects Lola enrichment on *Dref* or *SkpA* binding regions. These results suggest that Wts-Lola interaction and Lola binding to DNA are two independent events. Wts possibly interacts with Lola at the C-terminus in regions outside of the zinc finger motifs, structuring conformational changes which indirectly affect interaction between proteins and other parts of Lola such as the N-terminal BTB domain. It has been shown that BTB domain-containing proteins serve as both a linker and substrate adaptor within the Cul3-based E3 ligases (*Furukawa et al., 2003*; *Geyer et al., 2003*; *Pintard et al., 2004*). Proteins containing either the Leucine-rich repeats (LRR) or WD40 domain, such as the F-box protein in the SCF E3 complex, have been shown to mediate the degradation of BTB domain substrate adaptor (*Wimuttisuk et al., 2014*). Interestingly, our results indicate that Lola is degraded via a UPS-dependent mechanism and its ubiquitination is affected by the presence of Wts. Based on these findings, it is possible that Wts-Lola interaction modulates the interaction between Lola and other LRR or WD40-containing proteins, hence preventing Lola ubiquitination and degradation by the UPS system.

## Hippo-Lola signaling in normal midgut homeostasis

During stress or injury-induced regeneration, ISCs rapidly proliferate in order to replenish the cell loss in a short time (*Amcheslavsky et al., 2009*; *Buchon et al., 2009a*; *Buchon et al., 2010*; *Buchon et al., 2009b*). Abundant evidence has suggested that Yki is a central Hippo effector in the regeneration process. Not only that Yki protein levels and transcription levels of Hippo downstream target genes are upregulated during regeneration, loss of Yki in either precursors or ECs blocks DSS- or infection-induced ISC proliferation, respectively. *yki* overexpression also leads to activation of EGFR and JAK-STAT signaling pathways (*Karpowicz et al., 2010*; *Ren et al., 2010*; *Shaw et al., 2010*; *Staley and Irvine, 2010*). These results suggest that Yki and Hippo signaling are important for midgut regeneration.

Unlike regeneration, normal midgut homeostasis only requires a basal level of cell turnover and ISCs proliferate to a minimum extent to maintain the need (*Antonello et al., 2015*; *Jiang et al., 2011*). During normal condition, Hippo signaling restricts ISC proliferation, thus serving an inhibitory role in regulating ISC proliferation and midgut homeostasis. Inactivation of Wts or Hpo, similarly as *yki* overexpression, leads to inactivation of Hippo signaling, thus enhancing ISC proliferation (*Karpowicz et al., 2010*; *Ren et al., 2010*; *Shaw et al., 2010*; *Staley and Irvine, 2010*). Nonetheless, Yki inactivation does not affect ISC proliferation, raising the argument that Yki is not as prominently needed in midgut homeostasis as in regeneration. Conserved and similarly in mammals, depletion of Yki homologs YAP1 and/or TAZ protein in the intestine does not affect normal homeostasis, indicating that YAP1 (*Barry et al., 2013*; *Cai et al., 2010*; *Camargo et al., 2007*) and TAZ (*Azzolin et al., 2014*) might be dispensable under normal conditions. Our results that Lola mediates ISC proliferation via non-canonical Hippo signaling resolve the argument by supporting a complementary yet equally important role for Lola during Hippo-mediated ISC proliferation and midgut homeostasis. Upstream components might trigger Hippo signaling via controlling Wts protein levels instead of Wts phosphorylation, hence regulating Lola stability. Whereas Yki is critically needed during injury-induced regeneration, Lola plays a more fundamental and maintenance role during homeostasis, a process that Yki is dispensable. Collectively, our work uncovers a novel mechanism that Hippo regulates ISC proliferation via Lola-mediated non-canonical downstream signaling; Hippo-Lola signaling controls ISC basal proliferation and midgut homeostasis, whereas Yki in the canonical Hippo signaling controls rapid ISC proliferation during regeneration.

In the present study, we uncover a novel transcription factor Lola that functions downstream of Hippo signaling in regulating *Drosophila* midgut homeostasis. During normal homeostasis and maintenance, the Wts-Lola-Dref/SkpA signaling axis serves as a critical mediator restricting ISC proliferation, adding another layer of complexity in the regulatory mechanism of ISC proliferation. Considering the importance of Hippo signaling in intestinal diseases and tumorigenesis, our findings provide new insights on developing potential biomarkers or strategies of therapeutic targets for anti-cancer research.

## Materials and methods

**Key resources table**

| Reagent type (species) or resource | Designation | Source or reference | Identifiers | Additional information |
|---|---|---|---|---|
| Gene (*Drosophila melanogaster*) | *lola* | | FBgn0283521 | |
| Cell line (*Drosophila melanogaster*) | S2 | ATCC | ATCC CRL-1963 RRID:CVCL_Z232 | |
| Genetic reagent (*Drosophila melanogaster*) | UAS-*lola* RNAi | Vienna *Drosophila* Resource Center | VDRC12574 | |
| Genetic reagent (*Drosophila melanogaster*) | UAS-*lola* RNAi | Vienna *Drosophila* Resource Center | VDRC12573 | |
| Genetic reagent (*Drosophila melanogaster*) | UAS-*lola* RNAi | Fly Stocks of National Institute of Genetics | NIG12052 R-1 | |
| Genetic reagent (*Drosophila melanogaster*) | FRT$^{42D}$*lola*$^{5D2}$ | Bloomington *Drosophila* Stock Center, PMID: 8050351 | Bloomington #28266 | |
| Genetic reagent (*Drosophila melanogaster*) | UAS-*wts* RNAi | Vienna *Drosophila* Resource Center, PMID: 21666802 | VDRC9928 | |
| Genetic reagent (*Drosophila melanogaster*) | UAS-*wts* RNAi | Bloomington *Drosophila* Stock Center, PMID: 23989952 | Bloomington #34064 | |

*Continued on next page*

*Continued*

| Reagent type (species) or resource | Designation | Source or reference | Identifiers | Additional information |
|---|---|---|---|---|
| Genetic reagent (*Drosophila melanogaster*) | UAS-wts$^{EPG4808}$ | Bloomington *Drosophila* Stock Center, PMID: 21278706 | Bloomington #30099 | |
| Genetic reagent (*Drosophila melanogaster*) | FRT$^{82B}$wts$^{x1}$ | PMID: 7743921 | | |
| Genetic reagent (*Drosophila melanogaster*) | FRT$^{42D}$yki$^{B5}$ | PMID: 16096061 | | |
| Genetic reagent (*Drosophila melanogaster*) | UAS-*yki* RNAi | PMID: 16096061 | | |
| Genetic reagent (*Drosophila melanogaster*) | UAS-*yki* | PMID: 16096061 | | |
| Genetic reagent (*Drosophila melanogaster*) | FRT$^{19A}$sd$^{\Delta B1}$ | PMID: 18258485 | | |
| Genetic reagent (*Drosophila melanogaster*) | UAS-*sd* RNAi | PMID: 18258485 | | |
| Genetic reagent (*Drosophila melanogaster*) | UAS-*Dref* RNAi | Vienna *Drosophila* Resource Center | VDRC22209 | |
| Genetic reagent (*Drosophila melanogaster*) | UAS-*Dref* RNAi | Bloomington *Drosophila* Stock Center | Bloomington #31941 | |
| Genetic reagent (*Drosophila melanogaster*) | UAS-*SkpA* RNAi | Bloomington *Drosophila* Stock Center | Bloomington #32870 | |
| Genetic reagent (*Drosophila melanogaster*) | UAS-*SkpA* RNAi | Bloomington *Drosophila* Stock Center | Bloomington #32991 | |
| Genetic reagent (*Drosophila melanogaster*) | *MS1096-Gal4; UAS-Dicer2* | PMID: 10557210 | | |
| Genetic reagent (*Drosophila melanogaster*) | *esg-Gal4/UAS-GFP; tubGal80$^{ts}$* | PMID: 16340959 | | |
| Genetic reagent (*Drosophila melanogaster*) | *MyoIA-Gal4/UAS-GFP; tubGal80$^{ts}$* | PMID: 16340959 | | |
| Genetic reagent (*Drosophila melanogaster*) | *MyoIA-Gal4; tubGal80$^{ts}$* | a gift from Rongwen Xi, National Institute of Biological Sciences, Beijing, China | | |
| Genetic reagent (*Drosophila melanogaster*) | *Upd3-LacZ* | a gift from Rongwen Xi | | |
| Genetic reagent (*Drosophila melanogaster*) | *bantam-lacZ* | PMID: 18258485 | | |
| Genetic reagent (*Drosophila melanogaster*) | *Stat-GFP* | PMID: 17008134 | | |
| Genetic reagent (*Drosophila melanogaster*) | *hsflp[122]; act > CD2>Gal4; UAS-Dicer2, UAS-GFP* | PMID: 9428512 | | |
| Genetic reagent (*Drosophila melanogaster*) | UAS-*lola* | This paper | | generated by microinjection |
| Antibody | anti-Delta (mouse monoclonal) | DSHB | Cat# c594.9b, RRID:AB_528194 | IF (1:100) |
| Antibody | anti-Prospero (mouse monoclonal) | DSHB | Cat# Prospero (MR1A), RRID:AB_528440 | IF (1:2000) |
| Antibody | anti- Arm (mouse monoclonal) | DSHB | Cat# N2 7A1 ARMADILLO, RRID:AB_528089 | IF (1:500) |

*Continued on next page*

*Continued*

| Reagent type (species) or resource | Designation | Source or reference | Identifiers | Additional information |
|---|---|---|---|---|
| Antibody | anti-p-H3 (rabbit polyclonal) | Cell Signaling | Cat# 9701, RRID:AB_331535 | IF (1:2000) |
| Antibody | anti-dpERK (rabbit monoclonal) | Cell Signaling | Cat# 4370, RRID:AB_2315112 | IF (1:1000) |
| Antibody | anti-β-gal (rabbit polyclonal) | Thermo Fisher | Cat# A-11132, RRID:AB_221539 | IF (1:500) |
| Antibody | anti-Phalloidin-TRITC | PMID: 28242614 | | IF (1:20000) |
| Antibody | anti- Cleaved Caspase-3 (rabbit polyclonal) | Cell Signaling | Cat# 9661, RRID:AB_2341188 | IF (1:100) |
| Antibody | anti-Pdm1 | a gift from Xiaohang Yang, Zhejiang University, China | | IF (1:500) |
| Antibody | anti-Dref | a gift from Masamitsu Yamaguchi, Kyoto Institute of Technology, Japan | | IF (1:100) |
| Antibody | anti-Yki | PMID: 23999857 | | IF (1:100) produced by immunizing rabbits with the Yki peptide of amino acids 180–418 |
| Antibody | anti-Lola | This paper | | IF (1:100) produced by immunizing rabbits with the peptide of Lola amino acids 1–467 |

## Plasmids and cloning

The full length *lola* DNA fragment (*lola-RD*: 2247 bp) was amplified from *Drosophila* cDNA (BDGP DGC clone *LD28033*) by PCR. *LolaN* and *C* are truncated *lola* variants with 1–1110 bp and 1111–2247 bp, respectively. *Lola ΔZF1, ΔZF2,* and *ΔZF12* are truncated *lola* variants deleting 1435–1498 bp, 1525–1596 bp, and 1435–1596 bp, respectively. For expression in S2 cells and flies, the DNA fragments mentioned above were cloned in frame with the Flag-tag in the *pUAST-Fg* vector according to the standard protocols. The NLS and NES sequences were listed as follow:

> NLS: 5'-CCTAAGAAGAAGAGGAAGGTT-3'
> NES: 5'-CTTCAGCTACCACCGCTTGAGAGACTTACTCTT-3'

## *Drosophila* Stocks and Genetics

The fly stocks used in this study are listed in the Key Resources Table.

## Temperature-controlled expression

The experiment using *esg-Gal4/UAS-GFP; tubGal80ᵗˢ* and *MyoIA-Gal4/UAS-GFP; tubGal80ᵗˢ* were crossed and cultured at 18℃ to restrict Gal4 activity. The F1 adult flies were shifted to 29℃ for 5 days to induce transgene expression. 10 female adult flies at least were dissected for each genotype, followed by immunostaining, microscopy, and statistical analysis.

## MARCM clonal analysis

GFP positive mutant clones were generated using the MARCM system. Flies were crossed and raised at 25℃, F1 adult flies with designated genotypes were subjected to heat shock at 37℃ for 1 hr and then cultured at 25℃ or 18℃ for days specified before dissection. Clones of more than 10 midguts were analyzed in each group.

## Immunohistochemistry

All immunostaining experiments were done on midguts of female flies. The guts were dissected and fixed in 4% formaldehyde (Sigma) for 1 hr and washed three times in PBS supplemented with 0.1% Triton X-100 (PBS-T). The guts were incubated in the primary antibody diluted in PBS-T overnight at 4°C, followed by three washes with PBS-T. After incubation with a second antibody at room temperature for 1 hr, midguts were washed for 3 times and mounted in PBS/glycerol medium with DAPI. Primary antibodies used in this study are listed in the Key Resources Table.

## Microscopy and statistical analysis

Fluorescent microscopy was performed on a Leica LAS SP8 confocal microscope; confocal images were obtained using the Leica AF Lite system. Confocal images from the basal layer of the posterior midguts (sub-region R5a, https://flygut.epfl.ch/overview) where ISCs can be clearly visualized were taken under 40 × objective. Single layer image is shown. The quantification of p-H3$^+$ cell per guts was undertaken by counting the p-H3$^+$ cell numbers over the whole gut of indicated genotypes. The p-H3$^+$ cell number quantification data were statistically present as average with the standard error of mean (SEM) and p-values of significance is calculated by Student's T-test (tails = 2, Two-sample unequal variance): * is $p<0.05$, ** is $p<0.01$, *** is $p<0.001$, ns is no significance with $p>0.05$.

To quantify the expression of indicated protein in *Figure 5* and *Figure 6*, the intensity of the indicated proteins and GFP signal in the view region was analyzed using the Leica AF Lite system. For each group, 50 (GFP$^−$) of which were quantified as wild type and 50 (GFP$^+$) were quantified as *lola$^{5D2}$* or *wts$^{x1}$*. The quantification data were statistically present as average with the standard error of mean (SEM) and p-values of significance is calculated by Student's T-test (tails = 2, Two-sample unequal variance): * is $p<0.05$, ** is $p<0.01$, *** is $p<0.001$, ns is no significance with $p>0.05$.

## Real-time PCR

Total RNAs were extracted from 10 midguts of 5-day-old female flies or 20 wing discs from third-instar larvae with indicated genotypes using Trizol Reagent (Invitrogen), and the cDNAs were synthesized using ReverTra Ace synthesis kit (Toyobo). Real-time PCR was performed using the SYBR Green Real-time PCR Master Mix (Toyobo) reagent with the ABI7500 System. Results were repeated for three independent biological replicates. *RpL32* was used as a normalized control. The RT-PCR data were statistically present as average with the standard error of mean (SEM) and p-values of significance were calculated by Student's T-test (tails = 2, Two-sample unequal variance) in Excel: * is $p<0.05$, ** is $p<0.01$, *** is $p<0.001$, ns is no significance with $p>0.05$.

## Cell line

S2 *Drosophila* cells from ATCC Ref: CRL-1963 were used. The identity has been athenticated (STR profiling). No mycoplasm contamination.

## Cell culture, Transfection, Immunoprecipitation, Ubiquitination assay, Western blot and immunostaining

S2 cells were cultured in *Drosophila* Schneider's Medium (Invitrogen) with 10% fetal bovine serum, 100 U/ml of penicillin, and 100 mg/ml of Streptomycin. Plasmid transfection was carried out using X-tremeGENE HP DNA Transfection Reagent (Sigma) according to manufacturer's instructions. A *ubiquitin-Gal4* construct and the indicated constructs (cloned into *pUAST* expression vectors) were co-transfected for all transfection experiments. CHX (50 mg/ml, Sigma) was used to inhibit nascent protein synthesis. MG132 (50 mg/ml, Sigma) and MG-101 (50 mg/ml, Sigma) were used to inhibit the UPS activity. E64 (50 mM; Sigma) and Leupeptin (50 mM; Sigma) were used to inhibit lysosome function. Immunoprecipitation, Ubiquitination assay, Western blot and immunostaining were performed according to standard protocols as previously described (*Guo et al., 2013*; *Zhang et al., 2008*; *Zhang et al., 2006*).

## RNA interference in S2 *Drosophila* cells

For RNAi in S2 cells, primers were designed as follows: *Renilla*-dsRNA-F (use as control) (5′- taatac-gactcaatagggatgacgtvaaaagtttac −3′); *Renilla*-dsRNA-R (use as control) (5′-

taatacgactcaatagggagactacatccggtttacc −3'); *lola*-dsRNA-F (5'- taatacgactcaatagggatggatgacgat-cagcagtt −3'); *lola*-dsRNA-R (5'- taatacgactcaatagggcggctgccggtccgctggac −3').

PCR reactions were used as templates for in vitro RNA production (in vitro Transcription T7 kit, Takara), and RNAs were purified using isopropyl-alcohol. dsRNAs were then annealed (68°C for 10 min and 37°C for 30 min). To perform knockdown experiment, S2 cells were diluted into $1 \times 10^6$ cells ml$^{-1}$ with serum free medium for 1 hr starvation with 15 μg dsRNA.

## ChIP

ChIP assays were performed using S2 cells. Cells were cross-linked for 15 min at 37°C in 1 ml of 1% formaldehyde in PBS buffer. The cross-linking was stopped by adding Glycine to a final concentration of 0.125M. Next, fixed cells were washed for 10 min in 1 ml ice-cold PBS for three times. Cells were sonicated in 1 ml sonication buffer (50 mM Hepes-KOH, pH 7.5, 140 mM NaCl, 1 mM EDTA, pH 8.0, 1% Triton X-100, 1% sodium deoxycholate, 0.1% SDS, and proteinase cocktail) with a Bioruptor sonicator. The sonication yielded genomic DNA fragments with an average size of about 200 bp. After centrifugation, lysates were incubated with 4 μg antibody for 4 hr (or overnight); 40 μl protein A/G PLUS agarose (Santa Cruz) was then added and incubated for another 4 hr (or overnight) on a rotator at 4°C. Beads were washed three times with the ChIP wash buffer (0.1% SDS, 1% Triton X-100, 2 mM EDTA, pH 8.0, 150 mM NaCl, and 20 mM Tris-Cl, pH 8.0), and then washed again with ChIP final wash buffer (0.1% SDS, 1% Triton X-100, 2 mM EDTA, pH 8.0, 500 mM NaCl, and 20 mM Tris-Cl, pH 8.0). Genomic DNA was eluted with elution buffer (1% SDS and 100 mM NaHCO$_3$) at 65°C for 30 min. 5 M NaCl was added to a final concentration of 200 mM for further incubation at 65°C for 4 hr (or overnight). Then, 0.5 M EDTA and 20 mg/ml proteinase K were added until their final concentrations were 5 mM and 0.25 mg/ml, respectively. The resulted mixture was incubated at 55°C for 2 hr to digest the protein. Genomic DNA was purified with a DNA purification kit (QIAGEN) and subjected to high throughput sequencing or real-time PCR.

The ChIP-seq data were deposited in the Gene Expression Omnibus (accession number GSE136999). The list of top 500 peaks of Lola ChIP-seq profile is shown in *Supplementary file 1* in detail.

## Luciferase reporter assay

To generate the *Dref-luc-1* or *Dref-luc-2* reporter genes, 2 DNA segments covering Lola-binding region of *Dref* were subcloned between BglII and KpnI sites of the pGL3-Promoter vector (*Dref-luc-1*: −304 to +43, *Dref-luc-2*: −211 to −3), respectively. For *SkpA-luc-1* or *SkpA-luc-2*, 2 DNA segments covering Lola-binding region of *SkpA* were subcloned downstream of the luciferaser gene into the BamHI site of the pGL3-Promoter vector (*SkpA-luc-1*: +1384 to +1716, *SkpA-luc-2*: +1458 to +1604), respectively.

For luciferase reporter assays, S2 cells were transfected with indicated reporters and copia-renilla luciferase reporter constructs in 24 well plate together with *Fg-Lola* or *Fg-LolaΔZF12* expressing constructs. Cells were incubated for 48 hr after transfection. The reporter assay was performed using the Dual-Luciferase reporter assay system (Promega). Dual-Luciferase measurements were performed in triplicate using GloMax-Multi JR Single-Tube Multimode Reader. The DLR data were statistically present as average with the standard error of mean (SEM) and p-values of significance is calculated by Student's T-test (tails = 2, Two-sample unequal variance) in Excel: * is p<0.05, ** is p<0.01, *** is p<0.001, ns is no significance with p>0.05.

## RNA-seq and statistical analysis

The transcriptomes were generated by RNA sequencing (RNA-seq) analysis using RNAs isolated from adult guts expressing *wts* RNAi, *lola* RNAi or *yki* using *Actin*$^{ts}$. 30 midguts dissected from indicated genotypes incubated for 5 days at 29°C were dissected to extract total RNAs per sample for RNA-Seq experiment.

After assessing RNA quality with Agilent Bioanalyzer, mRNAs were enriched by poly-A pull-down from total RNA samples (3 ug mRNA per sample, RIN >7 required). Then, sequencing libraries constructed with Illumina TruSeq Stranded mRNA Sample Prep Kit were sequenced using Illumina HiSeq X ten. We sequenced about 26M paired-end 150 bp reads for each sample.

After quality check with fastqc (version 0.11.8), we mapped the reads to an index of *Drosophila melanogaster* reference genome (BDGP6) using Hisat2 (version 2.1.0) (*Kim et al., 2015*). Then we converted sam files to bam files with samtools (version 1.9; *Li et al., 2009*). The aligned reads were assigned to genes using annotations from Ensembl (*Drosophila_melanogaster*.BDGP6.94.gtf) and HTseq-count (version 0.11.2) (*Anders et al., 2015*). The differentially expressed genes (DEGs) were identified using R package DESeq2 (version 1.25.12) (*Love et al., 2014*). Compared with the control group, genes with padj (adjusted p-value, corrected p-value after Multiple Comparisons by method 'BH' which DESeq2 provided)<0.05 were defined as DEGs. The union of all the DEGs from 3 groups of samples emerged into a cellection of 2116 DEGs.

The heatmap of the DEGs and hierarchical cluster analysis were employed to examine the correlation of gene expression pattern from indicated samples. We converted gene counts to Z-score to center and scaled the numeric matrix by R generic function 'scale'. The R package ComplexHeatmap (version 1.20.0) was used to draw the heatmap of the DEGs' expression from indicated samples (*Gu et al., 2016*). We used euclidean method to calculate distance between these two vectors in Hierarchical cluster analysis, and ward.D2 is used as the agglomeration method.

The raw RNA-seq data was deposited in the Sequence Read Archive (accession number SRP220236). The expression of 2116 DEGs indicated by reads counts of the RNA-seq profiles are shown in *Supplementary file 2*.

## Acknowledgements

We would like to thank Jin Jiang, Rongwen Xi, Xiaohang Yang, Masamitsu Yamaguchi, Vienna Stock Center, Bloomington Stock Center, NIG, Developmental Studies Hybridoma Bank for supplying antibodies and fly stocks. We would like to thank Gang Wei for the generous help on high-throughput sequencing data analysis. This work is supported by National Key Research and Development Program of China (2017YFA0103601, 2019YFA0802001), National Natural Science Foundation of China (No. 31530043, No. 31625017, No. 31871039, and No. 31900551), 'Strategic Priority Research Program' of the Chinese Academy of Sciences (XDB19000000), Shanghai Leading Talents Program.

## Additional information

### Funding

| Funder | Grant reference number | Author |
| --- | --- | --- |
| Chinese Academy of Sciences | Strategic Priority Research Program XDB19000000 | Lei Zhang |
| National Natural Science Foundation of China | No. 31530043 | Lei Zhang |
| National Natural Science Foundation of China | No. 31625017 | Lei Zhang |
| National Natural Science Foundation of China | No. 31871039 | Margaret S Ho |
| National Natural Science Foundation of China | No. 31900551 | Jinhui Li |
| National Natural Science Foundation of China | 2017YFA0103601 | Lei Zhang |
| National Natural Science Foundation of China | 2019YFA0802001 | Lei Zhang |

The funders had no role in study design, data collection and interpretation, or the decision to submit the work for publication.

### Author contributions

Xue Hao, Conceptualization, Formal analysis, Validation, Investigation, Project administration; Shimin Wang, Resources, Data curation, Investigation; Yi Lu, Resources, Data curation, Validation, Methodology; Wentao Yu, Formal analysis; Pengyue Li, Resources, Investigation; Dan Jiang, Methodology;

Tong Guo, Wenqing Wu, Resources; Mengjie Li, Data curation; Jinhui Li, Resources, Funding acquisition; Jinjin Xu, Resources, Methodology; Margaret S Ho, Conceptualization, Formal analysis, Supervision, Funding acquisition; Lei Zhang, Conceptualization, Supervision, Funding acquisition

### Author ORCIDs
Margaret S Ho https://orcid.org/0000-0002-2387-7564
Lei Zhang https://orcid.org/0000-0003-2566-6493

### Decision letter and Author response
Decision letter https://doi.org/10.7554/eLife.47542.sa1
Author response https://doi.org/10.7554/eLife.47542.sa2

## Additional files
### Supplementary files
• Supplementary file 1. Details of the top 500 peaks from lola ChIP-seq analysis.

• Supplementary file 2. The relative expression of 2116 DEGs indicated by reads counts from the RNA-seq profiles.

• Transparent reporting form

### Data availability
Sequencing data have been deposited in GEO under accession code GSE136999, and SRA under accession code SRP220236. All data generated or analysed during this study are included in the manuscript.

The following datasets were generated:

| Author(s) | Year | Dataset title | Dataset URL | Database and Identifier |
|---|---|---|---|---|
| Xue Hao | 2019 | Wts/Lola/Yki-induced intestinal stem cell (ISC) overproliferation affects gene expression in fly midgut | https://trace.ncbi.nlm.nih.gov/Traces/sra/sra.cgi?study=SRP220236 | Sequence Read Archive, SRP220236 |
| Hao X, Yu W, Zhang L | 2019 | Genome-wide binding of Lola in S2 cells | https://www.ncbi.nlm.nih.gov/geo/query/acc.cgi?acc=GSE136999 | NCBI Gene Expression Omnibus, GSE136999 |

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
