## [Decision Letter]

**Acceptance summary:**

The role of canonical Hippo signaling in restricting intestinal stem cell (ISC) proliferation is well established, but whether and how the non-canonical Hippo signaling is involved in ISC basal proliferation is less well-defined. This paper breaks new ground in identifying Lola as a transcription factor that acts as a non-canonical Hippo pathway component to restrict ISC proliferation independently of Yki-Sd. The findings are significant and add to the current knowledge of non-canonical Hippo signaling and will be appreciated in the intestinal stem cell and Hippo signaling fields.

**Decision letter after peer review:**

Thank you for submitting your article "Lola regulates *Drosophila* adult midgut homeostasis via non-canonical Hippo signaling" for consideration by *eLife*. Your article has been reviewed by three peer reviewers, and the evaluation has been overseen by a Reviewing Editor and Utpal Banerjee as the Senior Editor.

The reviewers have discussed the reviews with one another and the Reviewing Editor has drafted this decision to help you prepare a revised submission.

The reviewers were generally enthusiastic for this manuscript. It breaks new ground in identifying Lola as a transcription factor in a non-canonical Hippo signaling pathway However there were concerns raised by all three and in particular reviewer 3 that need to be addressed in the full revision as noted below:

Summary:

The role of canonical Hippo signaling in restricting intestinal stem cell (ISC) proliferation is well established, but whether and how the non-canonical Hippo signaling is involved in ISC basal proliferation are less well-defined.

In this paper, Hao et al. identify transcription factor Lola that acts as a non-canonical Hippo pathway component to restrict intestinal stem cell (ISC) proliferation independently of Yki-Sd, restricting ISC proliferation cell autonomously in the ISC and also non-cell-autonomously in the EC. Lola is stabilized by Warts (Wts) through direct protein-protein interactions. Lola rescues the enhanced ISC proliferation caused by Wts reduction or deletion by suppressing *Dref* and *SkpA* expression. The findings are significant and add to the current knowledge of non-canonical Hippo signaling. The data are of high quality, obtained by elegant experiments and implicate Lola as a non-canonical Hippo signaling component maintaining fly midgut homeostasis. These novel findings will be appreciated in the intestinal stem cell and Hippo signaling fields.

However a number of major concerns were raised which need to be addressed. One concern relates to the biological significance of Lola in controlling Hippo pathway outputs. Another concern relates to the lack of experiments showing in vivo regulation, in the gut, of endogenous Lola by Wts.

Essential revisions:

The authors should address the two major issues identified by the reviewers summarized below and in the detailed major comments:

1) Interaction between Wts and Lola.

As the authors demonstrate, Warts stabilizes Lola independent of Warts' kinase activity. In this context, Lola's activity is largely dependent upon the protein levels of Warts rather than its activity. However, much work on the Hippo kinase cascade (Hippo on/off) shows that Warts activity is mainly regulated by phosphorylation/dephosphorylation, without changes in protein levels. Given this, physiologically, it seems that the upstream regulators in the Hippo pathway that control ISC proliferation probably don't act through Lola. What types of regulation do act through Lola? How does Wts regulates Lola stability? Is Lola ever seen in the cytoplasm or is Warts ever seen in the nucleus? Where does the biochemical interaction between Wrts and Lola take place?

Although it is recognized that the whole pathway cannot be resolved in a single paper, going a bit further in the molecular mechanism would increase the impact of the presented work. The authors should consolidate the data on the interaction between Wts and Lola to provide a mechanistic view and address the detailed comments of the reviewers on this point by performing the experiments asked on this. The transcriptional role of Lola should also be documented.

2) in vivo regulation of endogenous Lola by Wts.

The reviewers raise the need to provide more convincing data that Hippo signaling really regulates endogenous Lola in vivo. The experiments (Figures 5, 6) all rely on overexpressed Lola acting as a dominant repressor of proliferation, and the interpretation of the results could be wrong (Lola might repress anything in this format). The connection between Lola and Wts (or Hippo signaling in general) in this cellular context and that this is how Hippo regulates ISC proliferation should be clarified. The case will be strengthened by trying another approach such as the reverse – overexpress Wts and testing to see if it's suppressive effect can be bypassed with loss of Lola.

Reviewer #1:

I only have one major question for the authors. How does Wts regulates Lola stability?

Other comment:

In Figure 1B, D, E, Yki RNAi further increased cell proliferation caused by *wts* RNAi. The authors should provide some explanation.

Reviewer #2:

Several issues need to be addressed prior to publication and proper quantification are required for several figures. Moreover, some images need improvement and the Materials and methods section needs to be further developed.

Several conclusions are drawn from knock down experiments with no direct validation of the efficiency of the RNAi transgenes. Could the authors provide molecular data/references on this?

Subsection “Lola restricts ISC proliferation by suppressing *Dref* and *SkpA* expression levels”. Lola is supposed to work as a transcriptional repressor, hence the KD of its targets is expected to rescue the phenotype induced by lola KD. This is not the case for all of them, could the authors explain this result? Also, it is puzzling that one of the targets tested functionally shows ChIP enrichment in the 3'UTR (*SkpA*). A direct assay for the transcriptional control of Lola (at least on this gene) would validate the hypothesis of the authors, they could for example perform transfection assays in S2 cells.

Since a key message of the manuscript is that Wrts stabilizes Lola, Is Lola ever seen in the cytoplasm or is Warts ever seen in the nucleus? Where does the biochemical interaction between Wrts and Lola take place? Although the whole pathway cannot be resolved in a single paper, going a bit further in the molecular mechanism would increase the impact of the presented work.

Reviewer #3:

One major concern relates to the biological significance of Lola in controlling Hippo pathway outputs. As the authors demonstrate, Warts stabilizes Lola independent of Warts' kinase activity. In this context, Lola's activity is largely dependent upon the protein levels of Warts rather than its activity. However, much work on the Hippo kinase cascade (Hippo on/off) shows that Warts activity is mainly regulated by phosphorylation/dephosphorylation, without changes in protein levels. Given this, physiologically, it seems that the upstream regulators in the Hippo pathway that control ISC proliferation probably don't act through Lola. What types of regulation do act through Lola is not addressed in this paper. Another big problem is that there aren't really any good experiments showing in vivo regulation, in the gut, of endogenous Lola by Wts. The experiments (Figures 5, 6) all rely on overexpressed Lola acting as a dominant repressor of proliferation, and the interpretation of the results could be wrong (Lola might repress anything in this format). So, while the work on Lola is good, the connection between Lola and Wts (or Hippo signaling in general) in this cellular context is not demonstrated in a convincing way.

Specific points follow below:

1) Although knockdown of Yki in ISCs doesn't change the basal level of ISC proliferation, silencing Yki in ISCs does suppress the stress-induced proliferative response (Ren et al., 2010; Shaw et al., 2010). This indicates that Yki is essential in ISCs for regenerative proliferation. To demonstrate that Lola is functionally as relevant as Yki in ISCs, the authors could test if overexpression of Lola in ISCs also represses stress-induced proliferation (e.g. P.e infection or DSS treatment).

2) Although the genetic data shows that co-expression of *SkpA* or *Dref* RNAi in *esg^ts^*>*wts*RNAi midguts dramatically reduced the increase of ISC mitosis (Figure 6A), to solidly demonstrate that Warts-Lola-*Dref/SkpA* is a linear pathway, the authors need to sort out (FACS sorting technique) the GFP+-progenitor cells from the "*esg^ts^*GFP>*wts*RNAi" guts and run qPCR to check if *Dref/SkpA* are transcriptionally upregulated by Wts-KD.

3) Is overexpression of *Dref /SkpA* in ISCs (autonomous role) or ECs (non-autonomous role) sufficient to trigger ISC mitosis?

4) EGFR ligands and Upd1,2,3 are upregulated upon Lola-KD in ECs (Figure 2—figure supplement 2). These genes have been demonstrated to be regulated by the canonical Hippo/Yki pathway in ECs upon stress conditions. Given this, the authors should clarify if these genes' upregulation is Yki-dependent or not. In addition, the authors show that *esg^ts^*>LolaRNAi upregulates Bantam, which is a classical Yki target. So, what is the Yki's function in this context? Please explain.

5) Following the above question, it is well known that apoptotic ECs will generate Upd2/3 to promote gut regeneration. The authors need to clarify whether LolaRNAi in ECs promotes ISC mitosis through a specific signaling pathway, or just through triggering non-specific EC apoptosis (Myo1Ats> LolaRNAi, DCP-1 or Caspase-3 staining is recommended).

6) As noted above, the data in Figure 5 and 6 don't really establish that endogenous Lola is regulated by Wts in ISCs. This is a major shortcoming of the paper. This could be addressed by overexpressing Wts, to see if it can counteract Lola-RNAi to suppress ISC proliferation. If it can, well then Wts probably has other targets that regulate ISC proliferation. As noted above, it would also be helpful to show data on endogenous Lola protein (not detected in Figure 5I), and epistasis experiments where Lola DNA binding and effects on transcriptional targets were tested.

7) Figure 6A is a nice dataset that positions *Dref/SkpA* at the downstream of Wts. But, it would be better to perform these epistasis tests using Myo1Ats as well.

---

## [Author Response]

Essential revisions:The authors should address the two major issues identified by the reviewers summarized below and in the detailed major comments:1) Interaction between Wts and Lola.As the authors demonstrate, Warts stabilizes Lola independent of Warts' kinase activity. In this context, Lola's activity is largely dependent upon the protein levels of Warts rather than its activity. However, much work on the Hippo kinase cascade (Hippo on/off) shows that Warts activity is mainly regulated by phosphorylation/dephosphorylation, without changes in protein levels. Given this, physiologically, it seems that the upstream regulators in the Hippo pathway that control ISC proliferation probably don't act through Lola. What types of regulation do act through Lola? How does Wts regulates Lola stability? Is Lola ever seen in the cytoplasm or is Warts ever seen in the nucleus? Where does the biochemical interaction between Wrts and Lola take place?Although it is recognized that the whole pathway cannot be resolved in a single paper, going a bit further in the molecular mechanism would increase the impact of the presented work. The authors should consolidate the data on the interaction between Wts and Lola to provide a mechanistic view and address the detailed comments of the reviewers on this point by performing the experiments asked on this. The transcriptional role of Lola should also be documented.

To address these questions, new data have been added to the revised manuscript and details were given below point-by-point to each reviewer’s comment.

2) in vivo regulation of endogenous Lola by Wts.The reviewers raise the need to provide more convincing data that Hippo signaling really regulates endogenous Lola in vivo. The experiments (Figures 5, 6) all rely on overexpressed Lola acting as a dominant repressor of proliferation, and the interpretation of the results could be wrong (Lola might repress anything in this format). The connection between Lola and Wts (or Hippo signaling in general) in this cellular context and that this is how Hippo regulates ISC proliferation should be clarified. The case will be strengthened by trying another approach such as the reverse – overexpress Wts and testing to see if it's suppressive effect can be bypassed with loss of Lola.

Experiments using a second reverse approach by overexpressing *wts* in *lola* mutant clones have been done. Please see below for detailed responses and also revision in the manuscript.

Reviewer #1:I only have one major question for the authors. How does Wts regulates Lola stability?

Our evidence suggests that Wts interacts with Lola C-terminus. This interaction potentially causes conformational changes which indirectly affect interaction between proteins and other parts of Lola such as the N-terminal BTB domain. It has been shown that BTB domain-containing proteins serve as both a linker and substrate adaptor within the Cul3-based E3 ligases (Furukawa et al., 2003; Geyer et al., 2003; Pintard, Willems, and Peter, 2004). Proteins containing either the Leucine-rich repeats (LRR) or WD40 domain, such as the F-box protein in the SCF E3 complex, mediate the degradation of BTB domain substrate adaptor (Wimuttisuk et al., 2014). Interestingly, our results suggest that Lola is degraded via UPS-dependent (SCF complex-dependent) mechanism and that its ubiquitination is moderately abolished in the presence of Wts (Figure 6—figure supplement 1A-B in the revised manuscript). Based on these findings, it is possible that Wts-Lola interaction modulates the interaction between Lola and other LRR or WD40-containing proteins, hence preventing Lola ubiquitination and degradation by UPS.

Multiple upstream signals integrate to activate the Hippo pathway. Despite that Hippo signaling mainly transduces via triggering Wts phosphorylation (Udan et al., 2003; Wu et al., 2003), previous studies indicate that some upstream components regulate the Hippo signaling by controlling Wts protein levels. The atypical cadherin Fat (Ft) (Cho et al., 2006), the atypical myosin Dachs (D) together with the LIM domain protein Zyxin (Zyx) (Rauskolb et al., 2011), and the tumor suppressor gene Scribble (Scrib) (Verghese et al., 2012) function as components of the Hippo pathway via regulating Wts stability. It is possible that these upstream components signal through Wts protein level to control Lola stability for downstream ISC proliferation.

Regarding where the Wts-Lola interaction takes place, both Wts and Lola proteins have been predicted to contain the nuclear export signal (NES) (http://www.cbs.dtu.dk/services/NetNES/) (la Cour et al., 2004) and the nuclear localization signal (NLS) (http://nls-mapper.iab.keio.ac.jp/cgi-bin/NLS_Mapper_form.cgi) (Kosugi et al., 2009), respectively. These results indicate that Wts possibly localizes to the nucleus in addition to the cytoplasm, whereas Lola localizes to the cytoplasm in addition to the nucleus (Author response image 1). Our cell culture results indicate that Wts affects Lola stability both in the cytoplasm and the nucleus. An NLS or NES sequence was added to the *wts* 3’ end to drive Wts entry to or exit out of the nucleus, respectively. Western blot analysis indicated that the Wts^NLS^ expression caused a remarkable increase in Lola protein levels, suggesting that Wts^NLS^ stabilizes Lola in the nucleus (Figure 6—figure supplement 1C in the revised manuscript). Immunostaining results also revealed a diffuse pattern of Lola colocalizing with Wts^NLS^ in the nucleus (Figure 6—figure supplement 1D in the revised manuscript). In contrast, Wts^NES^ expression did not cause such colocalization pattern in the nucleus and only diminished the increase of Lola protein levels partially (Figure 6—figure supplement 1C, D in the revised manuscript), suggesting the possibility of Lola translocation to the cytoplasm where it is protected by Wts. Taken together, these results suggest that Wts interacts with Lola both in the cytoplasm and the nucleus.

These results have been added to the revised manuscript and discussed in the Discussion section.

Other comment:In Figure 1B, D, E, Yki RNAi further increased cell proliferation caused by wts RNAi. The authors should provide some explanation.

We thank the reviewer for the comments. To accurately analyze the difference on cell proliferation between *wts* RNAi alone and *wts* RNAi + *yki* RNAi, we increased the sample size for each genotype. A more detailed analysis showed that there is no statistical significance between results from these two samples. These new results were shown in Figure 1E in the revised manuscript.

Reviewer #2:Several issues need to be addressed prior to publication and proper quantification are required for several figures. Moreover, some images need improvement and the Materials and methods section needs to be further developed.

We thank the reviewer for the comments. New quantifications have been made for Figure 1F-H’, Figure 5D-E’, Figure 6G-H’’, and images in Figure 1F-H’, Figure 7G have been improved. The *Drosophila* Stocks and Genetics, Microscopy and Statistical Analysis, and RNA-seq and Statistical Analysis sections in the Materials and methods section have been revised. Please see the revised manuscript and figures, also throughout different sections in this response letter.

Several conclusions are drawn from knock down experiments with no direct validation of the efficiency of the RNAi transgenes. Could the authors provide molecular data/references on this?

The efficiencies of RNAi lines for *wts, lola, Dref*, and *SkpA* have been verified by qPCR. As shown in Author response image 2, the expression levels of these genes were efficiently repressed. We have added the references of several RNAi transgenes in the Key Resources Table in the Materials and methods section.

**Author response image 2. respfig2:** 

Subsection “Lola restricts ISC proliferation by suppressing Dref and SkpA expression levels”. Lola is supposed to work as a transcriptional repressor, hence the KD of its targets is expected to rescue the phenotype induced by lola KD. This is not the case for all of them, could the authors explain this result? Also, it is puzzling that one of the targets tested functionally shows ChIP enrichment in the 3'UTR (SkpA). A direct assay for the transcriptional control of Lola (at least on this gene) would validate the hypothesis of the authors, they could for example perform transfection assays in S2 cells.

To address reviewer’s concern, luciferase reporter assays have been performed. *lola* binding regions of *Dref* or *SkpA* (1: long, 2: short) were cloned into the reporter, among them the *SkpA* 3’UTR region was cloned to the C-terminus. Our results showed that transcription activities of these reporters were repressed in the presence of Lola, while the dual reporter activity reflecting Sd-Yki transcription *3xSd luc* (Zhang et al., 2008) was not affected. Expression of Lola lacking the DNA binding zinc finger regions (Fg-Lola∆ZF12) did not cause repression. These results showed that *Dref* and *SkpA* are under Lola transcriptional control and directly repressed by Lola. These results were shown in Figure 5C in the revised manuscript.

Since a key message of the manuscript is that Wrts stabilizes Lola, Is Lola ever seen in the cytoplasm or is Warts ever seen in the nucleus? Where does the biochemical interaction between Wrts and Lola take place? Although the whole pathway cannot be resolved in a single paper, going a bit further in the molecular mechanism would increase the impact of the presented work.

Similar questions have been raised by reviewer 1. Please see below for our response:

Regarding where the Wts-Lola interaction takes place, both Wts and Lola proteins have been predicted to contain the nuclear export signal (NES) (http://www.cbs.dtu.dk/services/NetNES/) and the nuclear localization signal (NLS) (http://nls-mapper.iab.keio.ac.jp/cgi-bin/NLS_Mapper_form.cgi), respectively. These results indicate that Wts possibly localizes to the nucleus in addition to the cytoplasm, whereas Lola localizes to the cytoplasm in addition to the nucleus (Author response image 1). Our cell culture results indicate that Wts affects Lola stability both in the cytoplasm and the nucleus. An NLS or NES sequence was added to the *wts* 3’ end to drive Wts entry to or exit out of the nucleus, respectively. Western blot analysis indicated that the Wts^NLS^ expression caused a remarkable increase in Lola protein levels, suggesting that Wts^NLS^ stabilizes Lola in the nucleus (Figure 6—figure supplement 1C in the revised manuscript). Immunostaining results also revealed a diffuse pattern of Lola colocalizing with Wts^NLS^ in the nucleus (Figure 6—figure supplement 1D in the revised manuscript). In contrast, Wts^NES^ expression did not cause such colocalization pattern in the nucleus and only diminished the increase of Lola protein levels partially (Figure 6—figure supplement 1C, D in the revised manuscript), suggesting the possibility of Lola translocation to the cytoplasm where it is protected by Wts. Taken together, these results suggest that Wts interacts with Lola both in the cytoplasm and the nucleus.

Reviewer #3:One major concern relates to the biological significance of Lola in controlling Hippo pathway outputs. As the authors demonstrate, Warts stabilizes Lola independent of Warts' kinase activity. In this context, Lola's activity is largely dependent upon the protein levels of Warts rather than its activity. However, much work on the Hippo kinase cascade (Hippo on/off) shows that Warts activity is mainly regulated by phosphorylation/dephosphorylation, without changes in protein levels. Given this, physiologically, it seems that the upstream regulators in the Hippo pathway that control ISC proliferation probably don't act through Lola. What types of regulation do act through Lola is not addressed in this paper. Another big problem is that there aren't really any good experiments showing in vivo regulation, in the gut, of endogenous Lola by Wts. The experiments (Figures 5, 6) all rely on overexpressed Lola acting as a dominant repressor of proliferation, and the interpretation of the results could be wrong (Lola might repress anything in this format). So, while the work on Lola is good, the connection between Lola and Wts (or Hippo signaling in general) in this cellular context is not demonstrated in a convincing way.Specific points follow below:1) Although knockdown of Yki in ISCs doesn't change the basal level of ISC proliferation, silencing Yki in ISCs does suppress the stress-induced proliferative response (Ren et al., 2010; Shaw et al., 2010). This indicates that Yki is essential in ISCs for regenerative proliferation. To demonstrate that Lola is functionally as relevant as Yki in ISCs, the authors could test if overexpression of Lola in ISCs also represses stress-induced proliferation (e.g. P.e infection or DSS treatment).

We thank the reviewer for the suggestion. *lola* overexpression in ISCs represses DSS-induced proliferation as shown by Author response image 3. This piece of evidence indicates that Lola is functionally as relevant as Yki in ISCs. However, despite the repression, the fact that *lola* overexpression artificially increases Lola protein levels might have led to repression in any format.

**Author response image 3. respfig3:** 

2) Although the genetic data shows that co-expression of SkpA or Dref RNAi in esg^ts^>wtsRNAi midguts dramatically reduced the increase of ISC mitosis (Figure 6A), to solidly demonstrate that Warts-Lola-Dref/SkpA is a linear pathway, the authors need to sort out (FACS sorting technique) the GFP+-progenitor cells from the "esg^ts^GFP>wtsRNAi" guts and run qPCR to check if Dref/SkpA are transcriptionally upregulated by Wts-KD.

According to the reviewer’s suggestion, the GFP^+^-progenitor cells were sorted by FACS analysis and qPCR analysis was conducted. As shown in Author response image 4, *esg^ts^* driven *wts* RNAi expression resulted in a knockdown efficiency of about 40% and only *Dref* mRNA levels was significantly upregulated. While results from the *wts^x1^* clonal guts in the manuscript exhibited a 50% reduction in *wts* expression, a more apparent increase in the *Dref/SkpA* mRNA levels was detected.

**Author response image 4. respfig4:** 

3) Is overexpression of Dref /SkpA in ISCs (autonomous role) or ECs (non-autonomous role) sufficient to trigger ISC mitosis?

According to the reviewer’s suggestion, the numbers of p-H3+ cells were analyzed when overexpressing *Dref/SkpA* in ISCs or ECs. As shown in Author response image 5, neither *Dref* nor *SkpA* overexpression is sufficient to activate ISC proliferation. It is possible that other factors downstream of Lola are required to function together with *Dref/SkpA* in regulating ISC proliferation.

**Author response image 5. respfig5:** 

4) EGFR ligands and Upd1,2,3 are upregulated upon Lola-KD in ECs (Figure 2—figure supplement 2). These genes have been demonstrated to be regulated by the canonical Hippo/Yki pathway in ECs upon stress conditions. Given this, the authors should clarify if these genes' upregulation is Yki-dependent or not. In addition, the authors show that esg^ts^>LolaRNAi upregulates Bantam, which is a classical Yki target. So, what is the Yki's function in this context? Please explain.

We thank the reviewer for the suggestion. mRNA levels of EGFR ligands and Upd1,2,3 were analyzed when expressing *lola* RNAi alone or *lola* + *yki* RNAi. Reducing *yki* expression partially decreased the elevated mRNA levels of these ligands (Author response image 6). We also examined the transcriptional level of microRNA *bantam*. As shown in Author response image 6’’’, elevated *bantam* expression levels upon Lola depletion were also partially affected, but not the expansion of ISC population. These results suggest that Yki might be moderately involved in EGFR and JAK-STAT signaling activation upon Lola depletion.

**Author response image 6. respfig6:** 

5) Following the above question, it is well known that apoptotic ECs will generate Upd2/3 to promote gut regeneration. The authors need to clarify whether LolaRNAi in ECs promotes ISC mitosis through a specific signaling pathway, or just through triggering non-specific EC apoptosis (Myo1Ats> LolaRNAi, DCP-1 or Caspase-3 staining is recommended).

We thank the reviewer for the suggestion. According to our results on Caspase-3 staining (Author response I mage 7), lacking *lola* in ECs does not trigger non-specific EC apoptosis.

**Author response image 7. respfig7:** 

6) As noted above, the data in Figure 5 and 6 don't really establish that endogenous Lola is regulated by Wts in ISCs. This is a major shortcoming of the paper. This could be addressed by overexpressing Wts, to see if it can counteract Lola-RNAi to suppress ISC proliferation. If it can, well then Wts probably has other targets that regulate ISC proliferation. As noted above, it would also be helpful to show data on endogenous Lola protein (not detected in Figure 5I), and epistasis experiments where Lola DNA binding and effects on transcriptional targets were tested.

We thank the reviewer for the comments. First, the effect of *wts* overexpression has been analyzed in *lola* MARCM clones (Figure 7—figure supplement 2 in the revised manuscript). Interestingly, *wts* overexpression does not suppress ISC proliferation induced by *lola* mutant, indicating that endogenous Lola is regulated by Wts in ISCs. Next, despite that it is difficult to detect endogenous Lola protein by WB analysis due to the nature of the antibodies, a faint but distinct band was detected in Figure 6J (long exposure). Furthermore, endogenous Lola protein levels decreased in *wts* mutant clones (Figure 6—figure supplement 3C-D’’’) or when *wts* RNAi is expressed (Figure 6G-I), indicating that endogenous Lola proteins are under Wts regulation.

Finally, *lola* transcription regulation on *Dref/SkpA* has been tested. Not only that target gene expression levels increased upon *lola* RNAi expression, direct repression was also detected using luciferase reporter assays. Furthermore, results from epistatic genetic interaction analysis showed that either *Dref* or *SkpA* RNAi expression rescues *lola* RNAi-induced ISC proliferation (Figure 5 and Figure 5—figure supplement 1 in the revised manuscript).

Taken together, these results demonstrate the endogenous Lola regulation by Wts in ISCs.

7) Figure 6A is a nice dataset that positions Dref/SkpA at the downstream of Wts. But, it would be better to perform these epistasis tests using Myo1Ats as well.

Thank you for the suggestion. We performed these epistasis experiments using *Myo1A^ts^*. Co-expression of *Dref* or *SkpA* RNAi dramatically inhibited the hyperproliferation induced by *wts* RNAi (Author response image 8). Together with results from experiments using *esg^ts^*, it is concluded that *Dref/SkpA* functions downstream of Wts both cell-autonomously and non-cell-autonomously. These results have been added to Figure 7—figure supplement 1E in the revised manuscript.

**Author response image 8. respfig8:** 

**References**

la Cour, T., Kiemer, L., Mølgaard, A., Gupta, R., Skriver, K., Brunak, S. (2004). Analysis and prediction of leucine-rich nuclear export signals

*Protein Eng. Des. Sel.,* 17(6):527-36

Kosugi S., Hasebe M., Tomita M., and Yanagawa H. (2009) Systematic identification of yeast cell cycle-dependent nucleocytoplasmic shuttling proteins by prediction of composite motifs. *Proc. Natl. Acad. Sci. USA* 106, 10171-10176